# AnyMod-LLVE: Low-Light Video Enhancement with Modality-Agnostic Inference

**Hangfeng Liang** [* 1 2]  **Yutao Hu** [* † 1 2]  **Yanhan Hu** [1]  **Xiaohan Wu** [1]  **Wenqi Shao** [3]  **Ying Fu** [4]

## Abstract

Low-light video enhancement (LLVE) remains a challenging task due to severe information degradation under low-illumination conditions. Recent multimodal approaches have significantly improved enhancement performance by incorporating auxiliary modalities, such as event streams and infrared images. However, these methods typically assume the availability of these modalities at inference, which is often not feasible in real-world scenarios. To solve this problem, in this work, we propose AMNet, a unified multimodal framework for LLVE, to support flexible modality-agnostic inference, where auxiliary modalities may be unavailable. To address the issue of modality absence, we introduce a Spatial-Spectral Dual-Gated Translator that learns the correspondence between auxiliary modalities and RGB inputs, producing implicit auxiliary representations to support the robust enhancement. Additionally, to fully facilitate the learning of cross-modal correspondence, we conduct large-scale multimodal pretraining based on the RGB-only dataset with synthetic auxiliary modalities. Extensive experiments demonstrate that AMNet could handle arbitrary inference-time modality combinations and exhibits superior performance for LLVE under modality absence conditions. Code and models are available on the project page.

## 1. Introduction

Low-light video enhancement (LLVE) aims to restore visually pleasing videos captured under insufficient illumi-

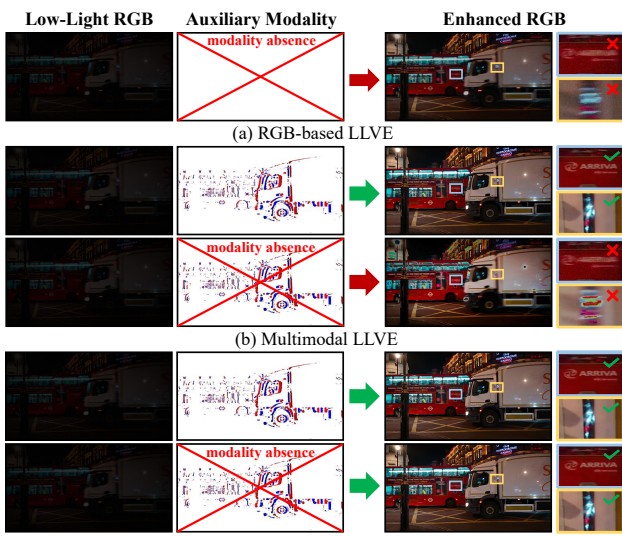

*Figure 1.* Comparison of different LLVE paradigms under missing-modality scenarios. **(a)** RGB-based LLVE relies solely on degraded RGB inputs and struggles to recover fine details. **(b)** Multimodal LLVE improves enhancement quality by leveraging auxiliary modalities (e.g., events), but fails when such modalities are missing at inference time. **(c)** Modality-agnostic LLVE enables robust enhancement by exploiting auxiliary modalities when available and maintaining stable performance when they are absent.

nation, and is crucial for real-world applications such as autonomous driving and intelligent surveillance. To this end, extensive efforts have been devoted to illumination modeling (Cai et al., 2023; Wang et al., 2025a; Wu et al., 2023) and temporal optimization (Wu et al., 2025; Xu et al., 2024; Fu et al., 2023), yielding significant improvements in visual fidelity and noise suppression. Nevertheless, as shown in Fig. 1(a), under extreme low-light conditions, RGB streams are severely information-limited, where critical cues such as object boundaries and fine structures are heavily corrupted or even missing. As a result, only RGB input cannot support reliable enhancement with faithful structure recovery. To mitigate this limitation, recent studies have incorporated auxiliary modalities, *e.g.*, event streams (Chen et al., 2025a; Kim et al., 2024) and infrared images (Wang et al., 2025b; Jha et al., 2025), to provide complementary temporal dynamics and structural priors, which have demonstrated great effectiveness in improving detail restoration for LLVE.

---
[*] Hangfeng Liang and Yutao Hu contribute equally. [1]School of Computer Science and Engineering, Southeast University [2]Key Laboratory of New Generation Artificial Intelligence Technology and Its Interdisciplinary Applications (Southeast University), Ministry of Education, China [3]Shanghai Innovation Institute [4]School of Computer Science and Technology, Beijing Institute of Technology. Correspondence to: Yutao Hu is the corresponding author <huyutao@seu.edu.cn>.

*Proceedings of the 43rd International Conference on Machine Learning*, Seoul, South Korea. PMLR 306, 2026. Copyright 2026 by the author(s).

However, recent multimodal LLVE methods still face practical limitations. As illustrated in Fig. 1(b), these approaches typically assume that auxiliary modalities are always available during both training and inference, resulting in a strong dependence on modalities such as event streams and infrared signals. However, since different sensors require additional hardware, careful calibration, and strict temporal/spatial synchronization, acquiring high-quality multimodal data is often costly in real-world applications. Consequently, modality absence issues often occurs, where auxiliary inputs are unavailable or partially corrupted. Unfortunately, existing multimodal LLVE models are not robust to such scenarios and suffer substantial performance drops, which limits the adaptability and deployability of multimodal LLVE methods in real-world scenarios.

To address the above issues, we propose **A**ny**M**od**Net**, a unified multimodal framework for LLVE that could support flexible modality-agnostic inference, particularly in the presence of missing modalities. As illustrated in Fig. 1(c), unlike prior approaches that treat auxiliary modalities as mandatory inputs at test time, AMNet models them as the implicit support that can be inferred from RGB information. Specifically, when auxiliary modalities are available, the framework effectively incorporates the explicit signals and extracts the structural information. On the other hand, when the auxiliary modalities are absent, AMNet can generate the specific implicit representation for the corresponding auxiliary modality from low-light RGB inputs, thereby preserving robust enhancement. Therefore, the proposed framework enables unified and robust inference across varying modality availability.

However, generating reliable implicit representations of auxiliary modalities solely from low-light RGB inputs is inherently challenging. Under extreme low illumination, critical fine-grained details, such as local textures and sharp edges, are severely fragile and often overwhelmed by sensor noise (Jiang et al., 2024), making it difficult to extract informative multimodal cues for these structures (Lu et al., 2025; Gasperini et al., 2023). To address this challenge, we propose a Spatial-Spectral Dual-Gated (S2DG) Translator, which explicitly leverages spectral analysis to extract reliable cues from degraded RGB features. Specifically, we first estimate an illumination distribution map and use an Illumination-Aware Detail Selector (IADS) to localize spatial regions that contain more reliable details. Then, we transform the features into the frequency domain and adaptively select the important bands via a Frequency-Band Selector (FBS) block. In this way, the S2DG Translator emphasizes the scarce yet informative details that survive in low-light observations, and preserves them as the key cues for implicit modality generation. Meanwhile, to retain global spatial context that may be suppressed by selective gating, we introduce a residual connection to keep the orig-

inal RGB feature content. This design enables the S2DG Translator to recover fine details from limited evidence without losing global structure and semantic consistency. Consequently, AMNet could synthesize high-quality implicit auxiliary representations even under severely degraded low-light conditions.

Furthermore, to facilitate the learning of cross-modal correspondence, we conduct large-scale pretraining on synthesized multimodal video data. Given the scarcity of paired multimodal LLVE datasets, we generate the pseudo multimodal data to enable the pretraining. Specifically, benefiting from recent advances in generative model, we can synthesize high-quality event streams (Hu et al., 2021) and infrared images (Xiao et al., 2025) conditioned on RGB inputs. Therefore, we leverage diverse video sources, such as datasets for video segmentation, video super-resolution, *etc*, as the basis to synthesize pseudo auxiliary modalities. Moreover, by adjusting the illumination condition, we create paired normal-light and low-light RGB inputs. Consequently, we can scale up the training data and pretrain the S2DG Translator to learn cross-modal correspondence as priors. Notably, although model-based synthesis can alleviate modality absence during training, invoking generative models at inference would introduce non-negligible latency. Therefore, the modality absence issue remains a practical problem at test time, highlighting the significance of the modality-agnostic inference.

Our main contributions are summarized as follows:

- We propose AMNet, a unified multimodal LLVE framework enabling modality-agnostic inference at test time. AMNet treats auxiliary modalities as optional cues rather than mandatory inputs, which can operate robustly under modality absence conditions.

- We introduce the S2DG Translator to synthesize implicit representations for auxiliary modality from low-light RGB, which distills scarce but informative cues from degraded RGB streams and strengthens cross-modal correspondence learning.

- Extensive experiments on various benchmark datasets demonstrate that AMNet achieves the state-of-the-art results for RGB-only LLVE. Moreover, AMNet remains highly robust when auxiliary modalities are absent at test time, exhibiting only a minimal performance drop compared to its multimodal setting.

## 2. Related Work

### 2.1. Low-Light Video Enhancement

For low-light video enhancement, most existing methods are developed under the RGB-only setting. Many methods

design enhancement strategies based on illumination decomposition or Retinex formulations (Wu et al., 2023; Cai et al., 2023). Other approaches focus on temporal optimization and consistency modeling to improve stability across frames (Wang et al., 2025a; Xu et al., 2024). Although these methods achieve noticeable improvements, they are fundamentally constrained by the limited and severely degraded information available in low-light RGB videos.

To alleviate this limitation, recent methods introduce auxiliary modalities to facilitate LLVE. Event streams (Chen et al., 2025a; Yao & Chuah, 2025; Sun et al., 2025) and infrared images (Wang et al., 2025b; Jha et al., 2025) provide complementary temporal dynamics and structural cues, leading to significant performance gains. However, most existing multimodal LLVE approaches require auxiliary modalities as mandatory inputs during inference, which substantially limits their applicability in practice. A recent attempt (Liu et al., 2023) explores handling modality absence via explicit auxiliary modality synthesis. Nevertheless, directly generating missing modalities at inference time typically involves substantial computational overhead and latency, making such solutions less practical for real-world deployment. Consequently, how to effectively exploit multimodal information during training while remaining robust to missing modalities at inference time remains an unsolved problem for LLVE.

## 2.2. Learning with Missing Modalities

Due to the need for additional hardware, careful calibration, and strict temporal/spatial synchronization across sensors, acquiring high-quality multimodal data is often costly and fragile in real-world applications. As a result, maintaining stable performance under partial modality availability has become an important problem.

A straightforward solution is to explicitly restore missing modalities from available ones via generative models (Dai et al., 2025; Kebaili et al., 2025). However, such approaches typically introduce substantial computational overhead and inference latency, which limits their practicality in time-sensitive scenarios. Alternatively, several works (Wu & Goodman, 2018; Dai et al., 2025; Liu et al., 2023) attempt to predict or synthesize implicit representations of missing modalities, providing auxiliary cues at inference time while reducing reliance on complete modality inputs. Beyond modality synthesis, another line of research focuses on improving model robustness under modality absence. These methods explore robust fusion mechanisms (Liaqat et al., 2025; Li et al., 2023; Alfasly et al., 2022) or cross-modal knowledge transfer (Wang et al., 2023b; Wei et al., 2023), enabling models to maintain stable performance under incomplete modality sets. Such strategies have shown effectiveness in applications including autonomous driving (Park

et al., 2025; Ge et al., 2023) and multimodal classification tasks (Park et al., 2023; Wang et al., 2023a).

Despite their success, most existing missing-modality learning methods assume relatively clean and well-structured of the available modalities, which is difficult to directly apply to LLVE tasks. Under extreme low-light conditions, RGB inputs suffer from severe noise and information degradation, substantially increasing the difficulty of producing reliable auxiliary modality representations.

Overall, while RGB-based LLVE methods have achieved notable progress through illumination modeling and temporal optimization, and multimodal approaches have demonstrated the benefits of auxiliary modalities, a unified solution that remains robust to modality absence scenarios during the inference time is still lacking. To bridge this gap, we propose **AMNet**, a unified LLVE framework that performs implicit modality generation at inference, enabling robust and modality-agnostic video enhancement under arbitrary modality availability.

## 3. Method

### 3.1. Overall Framework

Figure 2 illustrates the proposed AMNet, a unified multimodal framework for LLVE designed to support robust modality-agnostic inference. Given a low-light RGB video $\{R_t^{low}\}_{t=1}^T$, AMNet produces an enhanced video $\{R_t^{en}\}_{t=1}^T$. During training, auxiliary modalities such as event streams $\{\mathcal{E}_t\}_{t=1}^T$ and infrared images $\{I_t\}_{t=1}^T$ are available, while at inference time these modalities may be missing.

At each time step $t$, the low-light RGB frame $R_t^{low} \in \mathbb{R}^{H \times W \times 3}$ is first processed by an RGB encoder to extract multi-scale visual features $\mathcal{Z}_t^{rgb}$, which serve as the fundamental representations for subsequent enhancement and modality generation. When auxiliary modalities are available during training, they are processed by the corresponding modality encoders to extract modality features. Specifically, an event stream $\mathcal{E}_t = \{e_i\}_{i=1}^N$ is converted into an event voxel grid $E_t \in \mathbb{R}^{H \times W \times B}$ by assigning the polarity of each event to the two closest voxels, where $B$ denotes the number of voxels (Rebecq et al., 2019). The infrared image is represented as a single-channel image $I_t \in \mathbb{R}^{H \times W \times 1}$. These auxiliary modality features provide complementary structural information to RGB inputs.

To address the modality absence issue at inference time, AMNet incorporates a S2DG Translator that learns the correspondence between RGB inputs and auxiliary modalities, and extracts implicit auxiliary features from RGB features. When auxiliary modalities are unavailable, the generated features serve as supplements to support robust enhancement. Therefore, AMNet supports flexible modality-agnostic infer-

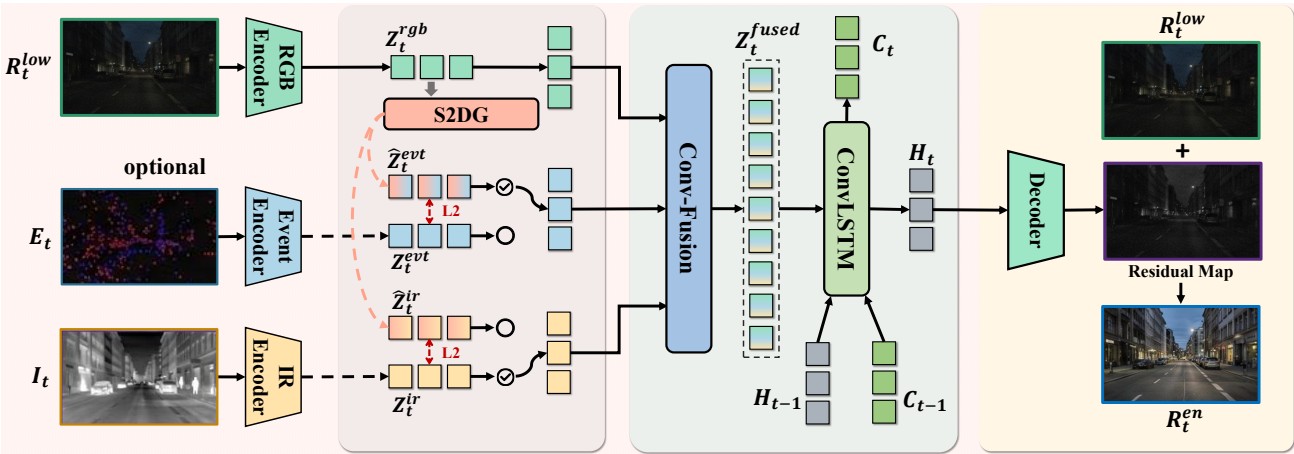

*Figure 2.* Overview architecture of the proposed AMNet. Notably, auxiliary modalities are optional, and the network remains robust under modality absence scenario via the synthetic implicit representation from S2DG Translator.

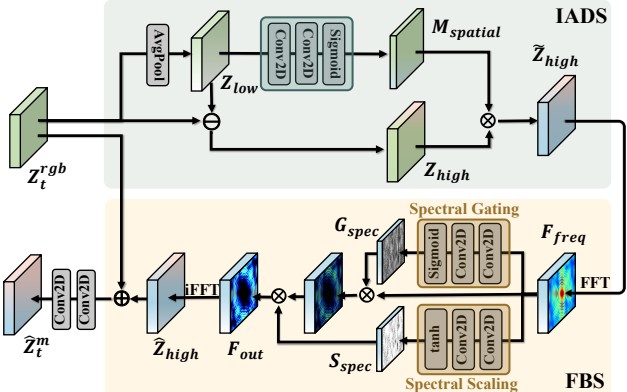

*Figure 3.* The details of the proposed S2DG Translator.

ence under arbitrary combinations of available modalities.

The RGB features and the auxiliary modality features are then fused and fed into a temporal modeling module to capture inter-frame dependencies. Finally, a decoder predicts a residual map, which will be combined with low-light input $R_t^{low}$ to form the final output $R_t^{en}$.

### 3.2. Spatial-Spectral Dual-Gated Translator

Auxiliary modalities such as event streams and infrared images provide critical high-frequency cues related to scene structure and motion, which are difficult to reliably recover from low-light RGB inputs. To address this challenge, we propose a Spatial-Spectral Dual-Gated (S2DG) Translator, which selectively emphasizes reliable high-frequency cues from degraded RGB features and progressively bridges the representation gap from RGB to auxiliary modality features. In this manner, S2DG enables the transfer of scarce yet informative fine-grained details from RGB observations into implicit auxiliary representations, which are often hard to identify under low-light conditions. By explicitly prioritiz-

ing these surviving detail cues, S2DG facilitates implicit modality generation with preserved structural information and facilitates cross-modal correspondence learning.

The architecture of the S2DG Translator is illustrated in Fig. 3, which consists of two components. First, an Illumination-Aware Detail Selector (IADS) evaluates the reliability of local spatial details based on illumination cues and suppresses noise-dominated regions. The resulting features are then processed by a Frequency-Band Selector (FBS) in the frequency domain, which selectively modulates informative spectral components to further strengthen cross-modal correspondence learning.

#### 3.2.1. ILLUMINATION-AWARE DETAIL SELECTOR

Under low-light conditions, high-frequency details in RGB features are highly sensitive to local illumination variations and are often dominated by noise in poorly illuminated regions, leading to inferior implicit modality representations (Chen et al., 2024b). To mitigate this issue, we introduce the IADS, which leverages low-frequency illumination information to identify and preserve reliable high-frequency details at each spatial location. Therefore, the scarce yet informative details are more effectively retained in the synthesized representations.

Inspired by (Chen et al., 2024a), we decompose $Z_t^{rgb}$ into a low-frequency component $Z_{low}$ and a high-frequency component $Z_{high}$ as:

$$\begin{cases} Z_{low} = \text{AvgPool}(Z_t^{rgb}), \\ Z_{high} = Z_t^{rgb} - Z_{low}, \end{cases} \quad (1)$$

where $Z_{low}$ captures global illumination distribution, while $Z_{high}$ indicates local detail responses mixed with noise. Based on $Z_{low}$, we predict an illumination-aware spatial reliability map to estimate the trustworthiness of high-frequency

details:

$$M_{spatial} = \sigma\big(\text{Conv}_{1\times 1}(Z_{low})\big), \qquad (2)$$

where $\sigma(\cdot)$ denotes the Sigmoid function. The high-frequency component is then spatially reweighted as:

$$\tilde{Z}_{high} = Z_{high} \odot M_{spatial}. \qquad (3)$$

Through this process, noise-dominated high-frequency details in poorly illuminated regions are suppressed.

### 3.2.2. FREQUENCY-BAND SELECTOR

In low-light scenarios, high-frequency details in RGB features are sparse and fragile due to severe noise interference, consisting of both attenuated structural signals and noise-dominated components. However, they serve as the key clues for transferring useful structural information from RGB to auxiliary modality representations. To this end, the S2DG Translator incorporates the FBS in the frequency domain, which selectively preserves and emphasizes informative frequency components that are critical for implicit modality generation, while suppressing unreliable, noise-dominated responses (Chen et al., 2025b).

Specifically, the processed feature $Z_{clean}$ is first transformed to the frequency domain:

$$F_{freq} = \mathcal{F}(\tilde{Z}_{high}), \qquad (4)$$

where $\mathcal{F}(\cdot)$ denotes a channel-wise 2D FFT operation. Based on $F_{freq}$, a spectral gating map $G_{\text{spec}}$ and a spectral scaling term $S_{\text{spec}}$ are predicted to filter reliable frequency components and adaptively enhance them:

$$\begin{cases} G_{\text{spec}} = \sigma\big(\text{Conv}(F_{freq})\big), \\ S_{\text{spec}} = \tanh\big(\text{Conv}(F_{freq})\big). \end{cases} \qquad (5)$$

Under the joint effect of gating and scaling, the frequency-domain features are modulated as:

$$F_{out} = F_{freq} \odot G_{\text{spec}} \odot (1 + S_{\text{spec}}). \qquad (6)$$

The modulated frequency-domain features are then transformed back to the spatial domain via inverse Fourier transform, yielding the enhanced high-frequency representation:

$$\hat{Z}_{high} = \mathcal{F}^{-1}(F_{out}). \qquad (7)$$

Finally, to compensate for the global spatial context that may be suppressed by selective gating, we fuse the low-frequency component $Z_{low}$ with the enhanced high-frequency features $\hat{Z}_{high}$ through a residual connection:

$$\hat{Z}_t^m = \hat{Z}_{high} + Z_{low}, \qquad (8)$$

The resulting feature $\hat{Z}_t^m$ serves as an implicit auxiliary modality representation for low-light video enhancement under modality-missing conditions. For different auxiliary modalities, we apply different S2DG Translators, enabling modality-specific implicit representation generation.

### 3.3. Optimization Objectives

Given the RGB features $Z_t^{rgb}$ together with the real infrared features $Z_t^{ir}$ and event features $Z_t^{evt}$, AMNet produces an enhanced frame via $\hat{R}_{t,en}^{full} = \mathcal{D}(Z_t^{rgb}, Z_t^{ir}, Z_t^{evt})$. Then, we define a reconstruction loss by measuring the difference between the enhanced frame and the corresponding normal-light reference frame $R_t^{gt}$:

$$\mathcal{L}_{rec}^{full}(\hat{R}_{t,en}^{full}, R_t^{gt}) = \mathcal{L}_p(\hat{R}_{t,en}^{full}, R_t^{gt}) + \lambda_s \mathcal{L}_s(\hat{R}_{t,en}^{full}, R_t^{gt}), \qquad (9)$$

where $\mathcal{L}_p$ denotes the pixel-wise $\ell_1$ loss and $\mathcal{L}_s$ denotes the structural similarity (SSIM) loss, which encourages both appearance fidelity and structural consistency with the reference frame.

To enhance robustness under modality absence, during training we additionally simulate modality absence scenarios and require AMNet to generate enhanced frames with incomplete auxiliary information. For example, when both event stream and infrared image are unavailable, AMNet produces the enhanced frame as $\hat{R}_{t,en}^r = \mathcal{D}(Z_t^{rgb}, \hat{Z}_t^{ir}, \hat{Z}_t^{evt})$. More generally, during training we consider all availability combinations of auxiliary modality $m \subset \{\text{ir}, \text{evt}\}$, including cases where only the event modality is available, only the infrared modality is available, or both auxiliary modalities are absent. For each setting $m$, AMNet produces a corresponding enhanced frame $\hat{R}_{t,en}^m$. Afterwards, for these enhanced frames, we also apply supervision via Eq. 9 as:

$$\mathcal{L}_{rec}^{miss} = \sum_{m \subset \{\text{ir}, \text{evt}\}} \mathcal{L}_{rec}^m(\hat{R}_{t,en}^m, R_t^{gt}). \qquad (10)$$

Meanwhile, to align the generated implicit modality representations with the real modality features, we introduce a feature-level distillation constraint. For each auxiliary modality $m \in \{\text{ir}, \text{evt}\}$, we minimize the discrepancy between the generated implicit representation $\hat{Z}_t^m$ and the corresponding real feature $Z_t^m$ of auxiliary modality input:

$$\mathcal{L}_{dt} = \sum_{m \in \{\text{ir}, \text{evt}\}} \lambda_m \left\| \frac{\hat{Z}_t^m}{\|\hat{Z}_t^m\|_2} - \text{sg}\left( \frac{Z_t^m}{\|Z_t^m\|_2} \right) \right\|_2^2, \quad (11)$$

where $\text{sg}(\cdot)$ denotes the "stop-gradient" operation to prevent gradients from flowing into the real-modality branch.

The overall training objective is defined as:

$$\mathcal{L}_{total} = \lambda_1 \mathcal{L}_{rec}^{full} + \lambda_2 \mathcal{L}_{rec}^{miss} + \lambda_3 \mathcal{L}_{dt}. \qquad (12)$$

## 4. Experiments

### 4.1. Experimental Setup

#### 4.1.1. DATASETS

**Pretraining data.** To ensure robust generalization under diverse low-light conditions, we collect multiple high-quality

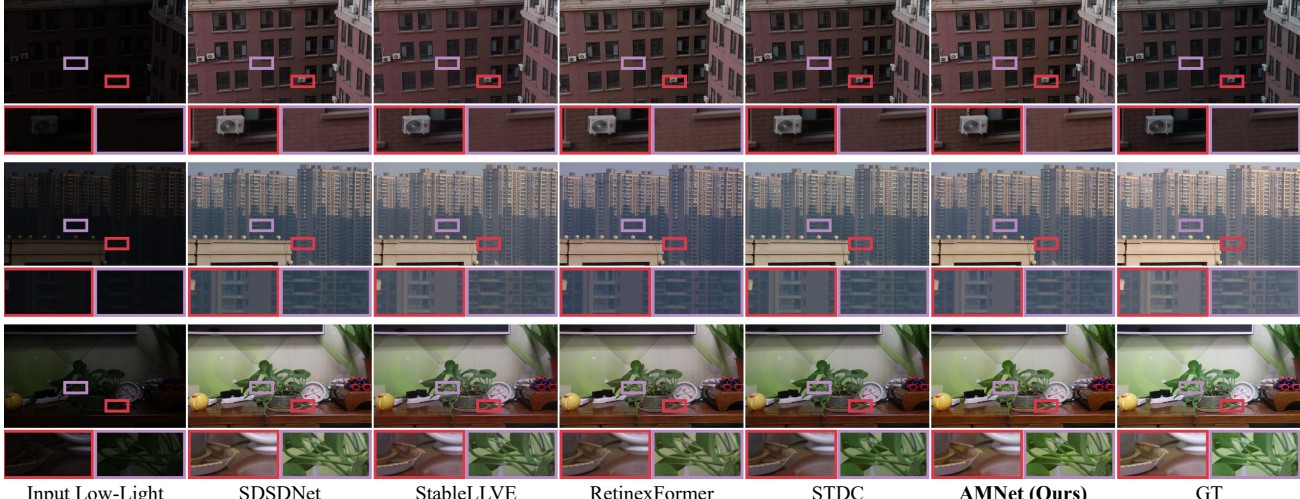

| Input Low-Light | SDSDNet | StableLLVE | RetinexFormer | STDC | **AMNet (Ours)** | GT |

*Figure 4.* Visual comparison on RGB only dataset (DID). Notably, in this experiment, our AMNet only takes RGB as input **without** the auxiliary modality.

public video datasets as sources for pretraining, covering diverse scene types, motion patterns, and content distributions. The statistics of these datasets are summarized in Table 7. To synthesize low-light inputs, we adopt the physics-based degradation model proposed in (Wei et al., 2020), where the illumination level is uniformly sampled within the range of 10%–50%, and additionally augmented with long-tail extreme low-light conditions sampled from 1%–10%. Meanwhile, we leverage v2e (Hu et al., 2021) and ThermalGen (Xiao et al., 2025) to generate synthetic event streams and infrared images from RGB videos, respectively, thereby constructing large-scale pseudo multimodal pretraining data.

**LLVE Datasets.** We evaluate our method on three widely used real-world LLVE datasets. **1)** DID (Fu et al., 2023) is a large-scale LLVE dataset captured under diverse illumination conditions using multiple cameras, containing 413 paired low-/normal-light videos with a total of 41,038 frames. Following (Xu et al., 2024), we split the dataset into training and testing sets with an 8:2 ratio. **2)** SDSD (Wang et al., 2021a) is a real-world LLVE dataset consisting of indoor (SDSD-indoor) and outdoor (SDSD-outdoor) subsets. We report the results on both subsets using the standard training/testing protocol. **3)** SDE (Chen et al., 2025a) is a real-world event-based LLVE dataset that provides paired low-/normal-light video sequences with synchronized event streams. It contains 43 outdoor and 48 indoor paired sequences, where each video consists of 145,888 frames.

#### 4.1.2. IMPLEMENTATION DETAILS

During training, input videos are divided into short clips of length 8. Video frames are randomly cropped to a resolution of $128 \times 128$, and random flipping is applied to improve

*Table 1.* Quantitative comparison with state-of-the-art methods under **RGB-only** setting. Notably, in this experiment, our AMNet only takes RGB as input **without** the auxiliary modality. **Red**: Best. **Blue**: Second best.

| Method | DID | | SDSD-Indoor | | SDSD-Outdoor | |
|---|---|---|---|---|---|---|
| | PSNR ↑ | SSIM ↑ | PSNR ↑ | SSIM ↑ | PSNR ↑ | SSIM ↑ |
| SMID | 22.28 | 0.84 | 24.84 | 0.72 | 23.30 | 0.67 |
| StableLLVE | 21.64 | 0.80 | 25.32 | 0.70 | 22.47 | 0.65 |
| SDSDNet | 22.52 | 0.81 | 26.81 | 0.75 | 23.08 | 0.71 |
| LLVE-CFA | 22.53 | 0.87 | 24.28 | 0.81 | 22.64 | 0.76 |
| RetinexFormer | 25.40 | 0.89 | 26.56 | 0.79 | 22.80 | 0.77 |
| BLLRVE | 23.25 | 0.81 | 26.02 | 0.74 | 23.50 | 0.71 |
| STCD | **30.10** | **0.93** | **28.93** | **0.88** | **26.32** | **0.82** |
| **Ours** | **31.57** | **0.95** | **29.03** | **0.92** | **26.37** | **0.84** |

training efficiency and data diversity. The model is trained using the AdamW optimizer with an initial learning rate of $2 \times 10^{-4}$, together with a cosine learning rate scheduler with 5 warm-up epochs. During multimodal pretraining, we perform distributed training on 4 NVIDIA A800 GPUs, and all downstream fine-tuning are conducted on a single NVIDIA A800 GPU with a batch size of 32.

**It is important to emphasize that**, to enable the fair comparison, for experiments on RGB-only dataset, AMNet only utilizes RGB streams as input on both fine-tuning and inference, without any auxiliary modality for supplement.

#### 4.2. Comparison with State-of-the-Art Methods

**Comparisons on RGB-only LLVE Datasets.** We fine-tune and evaluate our model on the DID and SDSD datasets using only RGB inputs. Our approach is compared with representative LLVE methods, including SMID (Chen et al., 2019), StableLLVE (Zhang et al., 2021), SDSDNet (Wang et al., 2021a), LLVE-CFA (Chhirolya et al., 2022), Retinex-

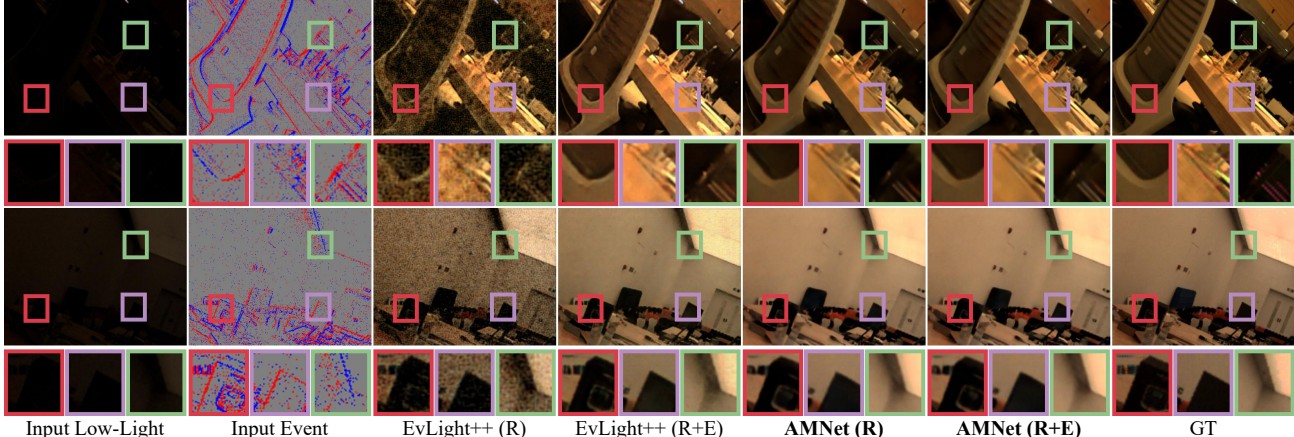

| Input Low-Light | Input Event | EvLight++ (R) | EvLight++ (R+E) | **AMNet (R)** | **AMNet (R+E)** | GT |

*Figure 5.* Visual comparison with multimodal LLVE methods on the SDE dataset under different modality availability. When auxiliary modalities are absent at inference (R), existing multimodal methods (*e.g.*, EvLight++) exhibit noticeable performance degradation. In contrast, AMNet maintains stable enhancement quality and preserves fine-grained details.

*Table 2.* Comparison with multimodal LLVE methods on SDE.

| Method | Inference Modality | SDE-Indoor | | SDE-Outdoor | |
|---|---|---|---|---|---|
| | | PSNR↑ | SSIM↑ | PSNR↑ | SSIM↑ |
| eSL-Net | R+E | 21.25 | 0.728 | 22.42 | 0.719 |
| ELIE | R+E | 19.98 | 0.617 | 20.69 | 0.653 |
| EvLowLight | R+E | 20.57 | 0.622 | 22.04 | 0.649 |
| LLVE-SEG | R+E | 21.79 | 0.705 | 22.35 | 0.690 |
| EvLight | R+E | 22.44 | 0.770 | 23.21 | 0.751 |
| EvLight++ | R+E | 22.67 | 0.779 | 23.34 | 0.768 |
| **Ours** | R | 23.04 | 0.816 | 23.75 | 0.775 |
| | R+E | **23.22** | **0.827** | **23.88** | **0.791** |
| | R+I | 23.12 | 0.822 | 23.81 | **0.778** |
| | R+E+I | **23.25** | **0.828** | **23.91** | **0.791** |

*Table 3.* Zero-shot performance comparison with **restoration foundation** model on LLVE benchmarks.

| Method | DID | | SDSD-Indoor | | SDSD-Outdoor | |
|---|---|---|---|---|---|---|
| | PSNR ↑ | SSIM ↑ | PSNR ↑ | SSIM ↑ | PSNR ↑ | SSIM ↑ |
| SGZ | **21.25** | **0.84** | 16.84 | 0.63 | 14.11 | 0.31 |
| Zero-TIG | 19.69 | 0.80 | 16.07 | 0.64 | 14.89 | 0.43 |
| FoundIR | 13.28 | 0.61 | 10.34 | 0.57 | 13.70 | 0.39 |
| MobileIE | 16.03 | 0.74 | 16.42 | 0.71 | **16.10** | 0.48 |
| DarkIR | 19.62 | 0.82 | **20.21** | **0.83** | 15.47 | **0.52** |
| **Ours** | **25.07** | **0.93** | **22.27** | **0.87** | **21.43** | **0.74** |

Former (Cai et al., 2023), BLLRVE (Zhang et al., 2024), and STCD (Xu et al., 2024). Reconstruction quality and structural consistency are evaluated using PSNR and SSIM, where higher values indicate better performance (Wang et al., 2004). As reported in Table 1, under this RGB-only setting, our method achieves the best performance on both datasets. On DID, PSNR and SSIM are improved by **1.47** dB and **0.02**, respectively, over the previous best method. On SDSD, PSNR / SSIM gains of **0.10** dB / **0.04** are obtained on the indoor subset, and **0.05** dB / **0.02** on the outdoor subset. These results indicate that AMNet exhibits robust and remarkable performance even without auxiliary modalities, validating its effectiveness under RGB-only inference.

Fig. 4 presents qualitative comparisons between AMNet and representative baselines on DID dataset. As shown, under low-light conditions, existing methods tend to increase overall brightness but often fail to recover fine-grained detail structures. In contrast, benefiting from the supplementary information from implicit representation of auxiliary modality, our method preserves sharper edges and more consistent

details with RGB-only training and inference.

**Comparisons on Multimodal LLVE Datasets.** We further compare our method with representative multimodal LLVE approaches on the SDE dataset (Liang et al., 2024), including eSL-Net (Wang et al., 2020), ELIE (Jiang et al., 2023), EvLowLight (Liang et al., 2023), LLVE-SEG (Liu et al., 2023), EvLight (Liang et al., 2024), and EvLight++ (Chen et al., 2025a). The SDE dataset is a multimodal LLVE dataset, contains real captured event streams. Quantitative results are summarized in Table 2. Notably, AMNet achieves the best performance even when only RGB inputs are available at inference, whereas competing methods rely on event streams. When event modalities are provided, AMNet further improves enhancement quality by effectively leveraging multimodal cues. Moreover, to investigate the scalability of AMNet under richer modality availability, we additionally synthesize infrared images using Thermal-Gen (Xiao et al., 2025). As shown in Table 2, even with synthesized (non-real) infrared inputs, AMNet consistently benefits from the additional modality, demonstrating further performance gains. These results highlight the potential value of our framework in real-world applications. The corresponding qualitative comparisons are provided in Fig. 5.

*Table 4.* Effect of pretraining scale for LLVE performance on RGB-only dataset, along with the distance between real and synthesized representations of auxiliary modality.

| Data Scale | DID | | | | SDSD-Indoor | | | | SDSD-Outdoor | | | |
|---|---|---|---|---|---|---|---|---|---|---|---|---|
| | PSNR ↑ | SSIM ↑ | L2(Event) ↓ | L2(IR) ↓ | PSNR ↑ | SSIM ↑ | L2(Event) ↓ | L2(IR) ↓ | PSNR ↑ | SSIM ↑ | L2(Event) ↓ | L2(IR) ↓ |
| 0% | 29.78 | 0.94 | 0.328 | 0.314 | 27.96 | 0.91 | 0.226 | 0.224 | 26.07 | 0.82 | 0.219 | 0.154 |
| 30% | 30.84 | 0.94 | 0.312 | 0.298 | 28.21 | 0.92 | 0.214 | 0.211 | 26.11 | 0.82 | 0.206 | 0.149 |
| 60% | 31.02 | 0.95 | 0.304 | 0.291 | 28.95 | 0.92 | 0.208 | 0.206 | 26.25 | 0.83 | 0.210 | 0.148 |
| 100% | **31.57** | **0.95** | **0.289** | **0.277** | **29.03** | **0.92** | **0.198** | **0.196** | **26.37** | **0.84** | **0.203** | **0.145** |

*Table 5.* Ablation study of Spatial-Spectral Dual-Gated Translator.

| IADS | FBS | DID | | SDSD-Indoor | | SDSD-Outdoor | |
|---|---|---|---|---|---|---|---|
| | | PSNR ↑ | SSIM ↑ | PSNR ↑ | SSIM ↑ | PSNR ↑ | SSIM ↑ |
| ✗ | ✗ | 29.85 | 0.93 | 28.31 | 0.91 | 25.93 | 0.81 |
| ✓ | ✗ | 30.30 | 0.93 | 28.60 | 0.91 | 26.05 | 0.81 |
| ✗ | ✓ | 30.95 | 0.94 | 29.20 | 0.92 | 26.25 | 0.82 |
| ✓ | ✓ | **31.57** | **0.95** | **29.03** | **0.92** | **26.37** | **0.84** |

**Zero-shot Performance.** We compare our approach with representative foundation models for restoration task, including SGZ (Zheng & Gupta, 2022), Zero-TIG (Li & Anantrasirichai, 2025), FoundIR (Li et al., 2025), MobileIE (Yan et al., 2025), and DarkIR (Feijoo et al., 2025). As shown in Table 3, without any fine-tuning, our method consistently outperforms existing foundation models across all datasets in the zero-shot manner. These results indicate that large-scale multimodal pretraining significantly enhances the generalization capability. Meanwhile, the implicitly auxiliary modality representations provide informative clues, thereby facilitating robust low-light enhancement in a zero-shot setting.

### 4.3. Ablation Studies

**Effect of Multimodal Pretraining.** During pretraining, the model is exposed to multiple modalities, which enables it to learn correspondence between RGB inputs and auxiliary modalities. To evaluate the effect of multimodal pretraining, we fix the downstream fine-tuning setting by using RGB-only inputs, and evaluate the performance under different pretraining data scales. Specifically, we vary the scale of multimodal data used in pretraining across {0%, 30%, 60%, 100%}. Besides PSNR and SSIM, we employ L2-distance to measure the discrepancy between real and synthesized auxiliary modality representations. As shown in Table 4, the RGB-only performance consistently improves across all benchmarks as the scale of multimodal pretraining data increases. At the same time, the discrepancy between the generated modality features and the real modality features decreases. Figure 6 visualizes the real and generated modality features under the 100% pretraining setting. These results indicate that large-scale multimodal pretraining facilitates

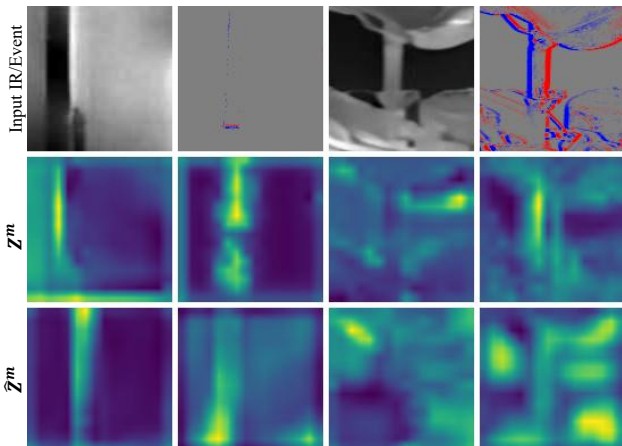

*Figure 6.* Visualization of real and synthetic feature representation for auxiliary modalities feature maps.

the learning of cross-modal correspondence, bring more realistic synthesized representations for missing modality and thereby improves the performance.

**Ablation on Model Components.** We investigate the contribution of the key components in the proposed Spatial-Spectral Dual-Gated (S2DG) Translator. Table 5 presents ablation results by selectively enabling the Illumination-Aware Detail Selector (IADS) and the Frequency-Band Selector (FBS). Both components individually improve performance. Combining IADS and FBS achieves the best performance across all benchmarks, validating their complementary roles in suppressing noise-dominated responses and preserving informative cues for implicit modality generation.

## 5. Conclusion

In this paper, we propose AMNet, a unified multimodal framework for LLVE that supports modality-agnostic inference. Unlike existing multimodal LLVE methods that rely on auxiliary modalities at test time, AMNet maintains remarkable performance under RGB-only inference. By incorporating the Spatial-Spectral Dual-Gated (S2DG) Translator, the model generates implicit auxiliary representations from severely degraded RGB inputs, enabling robust enhancement under missing-modality conditions. Moreover,

large-scale multimodal pretraining facilitates effective cross-modal correspondence learning, leading to strong zero-shot generalization and consistent gains during subsequent RGB-only fine-tuning. Extensive experiments demonstrate that AMNet achieves state-of-the-art performance under RGB-only inference settings, with further improvements when auxiliary modalities are available.

## Acknowledgements

This work was supported in part by the National Natural Science Foundation of China under Grant 82441021 and 62331006, in part by the Start-up Research Fund of Southeast University under Grant RF1028624156, and in part by the Big Data Computing Center of Southeast University.

## Impact Statement

This work aims to advance the field of low-light video enhancement by improving robustness and flexibility under varying modality availability. By leveraging multimodal supervision during training while supporting RGB-only inference, the proposed approach contributes to more reliable visual enhancement in challenging real-world conditions. Potential societal impacts include improved performance of vision systems in low-light environments, such as video capture, surveillance, and autonomous perception. We do not foresee significant negative societal consequences arising directly from this work.

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

## A. Appendix Outline

In this supplementary material, we provide more details of our method as follows:

- Sec. B presents detailed network architecture and implementation details of AMNet.

- Sec. C provides additional details of multimodal pretraining datasets.

- Sec. D analyzes the effectiveness of the proposed Spatial–Spectral Dual-Gated (S2DG) Translator.

- Sec. E shows additional qualitative results under the zero-shot setting.

- Sec. F studies robustness under different training and inference-time modality availability.

## B. Network Architecture and Implementation Details

Our framework adopts a RetinexFormer-based(Cai et al., 2023) RGB backbone. Specifically, we use an illumination-guided encoder–decoder architecture where the basic building unit is an illumination-guided attention block, and an illumination estimation branch predicts both illumination features and an illumination map to modulate multi-scale representations. To support multimodal training and inference, we extend the illumination estimation to a multimodal setting, allowing optional auxiliary inputs (event / IR) to provide additional cues for illumination guidance when available.

For auxiliary modalities, we employ lightweight modality encoders that directly downsample inputs to a compact latent representation at $H/8$ resolution, which substantially reduces computational overhead. When auxiliary modalities are missing, their latent features are produced by our S2DG Translator (described in Sec. D), which enables flexible combinations of real and generated modalities at test time.

For multimodal fusion, we incorporate an SNR-guided fusion strategy inspired by EvLight++(Chen et al., 2025a). An SNR map is estimated from the illumination-corrected RGB signal and used to adaptively weight RGB and auxiliary latent features, and fusion is performed only at the latent space ($H/8$) for efficiency. Temporal consistency is modeled using a ConvLSTM on the fused latent features, and the decoder predicts a residual map that is added to the input frame to obtain the final enhanced output. Since the full implementation is publicly released, we keep this section concise and summarize key reproducibility settings below. The detailed experimental configurations are summarized in Table 6.

*Table 6.* Experimental settings.

| Key | Value | Key | Value |
|---|---|---|---|
| Encoder channels | [64, 128, 256, 256] | Latent dimension | 256 |
| Clip length | 8 | Batch size | 32 |
| Clip stride | 2 | Training epochs | 100 |
| Crop size | $128 \times 128$ | Optimizer | AdamW |
| Event bins | 10 | Learning rate | $2 \times 10^{-4}$ |
| Warmup epochs | 5 | Weight decay | $1 \times 10^{-4}$ |
| Minimum learning rate | $1 \times 10^{-6}$ | Scheduler | cosine |
| EMA decay | 0.999 | $\lambda_1$ | 0.6 |
| $\lambda_2$ | 1.0 | $\lambda_3$ | 0.5 |
| $\lambda_s$ | 0.1 | $\lambda_{ir}$ | 0.5 |
| $\lambda_{evt}$ | 0.2 | Distillation warmup | 5 epochs |

## C. More Details of Multimodal Pretraining Datasets

As summarized in Table 7, we collect a diverse set of large-scale video datasets for multimodal pretraining, each contributing complementary scene characteristics and motion patterns. Together, these datasets provide video content at the scale of millions of frames, forming a large and diverse pretraining corpus that supports learning robust spatio-temporal and

cross-modal representations for large-scale LLVE. When constructing training clips, we adopt dataset-specific temporal strides to account for differences in frame rate and temporal redundancy across datasets.

Vimeo90K(Xue et al., 2019) is a large-scale dataset originally proposed for video restoration tasks such as super-resolution and frame interpolation, consisting of numerous short clips with natural scenes and smooth motion. It serves as a core source of generic spatio-temporal supervision, and clips are sampled following the dataset-specific temporal slicing configuration.

REDS(Nah et al., 2019) is a high-definition video dataset designed for benchmarking video restoration under challenging conditions such as blur, with high-quality frames and relatively stable motion. Clips from REDS are sampled using a temporal stride of 4.

YouTube-VOS(Xu et al., 2018) is a large-scale video object segmentation dataset featuring object-centric scenes with frequent motion, occlusion, and appearance changes. To preserve fine-grained object motion and temporal continuity, clips from YouTube-VOS are sampled with a shorter temporal stride of 2.

Inter4K(Stergiou & Poppe, 2022) consists of ultra-high-resolution videos with high frame rates (e.g., 60 fps) and rich visual details. To reduce excessive temporal redundancy caused by high-frame-rate sampling, clips from Inter4K are constructed using a larger temporal stride of 8.

UVO(Wang et al., 2021b) focuses on videos containing multiple interacting objects and complex dynamics. To balance motion coverage and redundancy in such crowded scenes, clips from UVO are sampled with a temporal stride of 4.

MOSEv2(Ding et al., 2025) contains complex scenes with diverse object interactions and motion patterns, providing additional diversity in temporal dynamics and scene composition. Clips from MOSEv2 are also sampled with a temporal stride of 4.

*Table 7.* Statistics of datasets used for multimodal pretraining.

| Dataset | Scene | #Videos | #Frames |
|---|---|---|---|
| Vimeo90K (Xue et al., 2019) | General | 91,701 | 641,907 |
| REDS (Nah et al., 2019) | HD | 203 | 20,220 |
| YouTube-VOS (Xu et al., 2018) | Object | 4,519 | 123,467 |
| Inter4K (Stergiou & Poppe, 2022) | 4K | 1,000 | 299,774 |
| UVO (Wang et al., 2021b) | Multi-obj | 1,023 | 295,823 |
| MOSEv2 (Ding et al., 2025) | Complex | 3,666 | 311,843 |
| Total | – | 102,112 | 1,693,034 |

## D. Effectiveness of the S2DG Translator

In this section, we analyze the effectiveness of the proposed S2DG Translator from both mechanism interpretability and generation quality. Specifically, we first visualize the spatial and spectral gating behavior to examine whether the translator selectively activates informative regions and channels. We then evaluate the quality of the implicitly generated modalities by comparing their feature distributions and downstream enhancement effects with those of real modalities.

### D.1. Visualization of Spatial and Spectral Gating

The S2DG Translator applies a two-stage gating mechanism to regulate high-frequency information during modality generation. A spatial filtering gate suppresses unreliable high-frequency responses in severely under-exposed regions. Frequency-domain gating then selects effective high-frequency bands and suppresses uninformative ones. Fig. 7 visualizes the spatial filtering maps and frequency-band gating responses.

### D.2. Implicit Modality Generation Quality

We evaluate the quality of implicitly generated modalities by comparing them with real modalities in feature space and in final enhancement results.

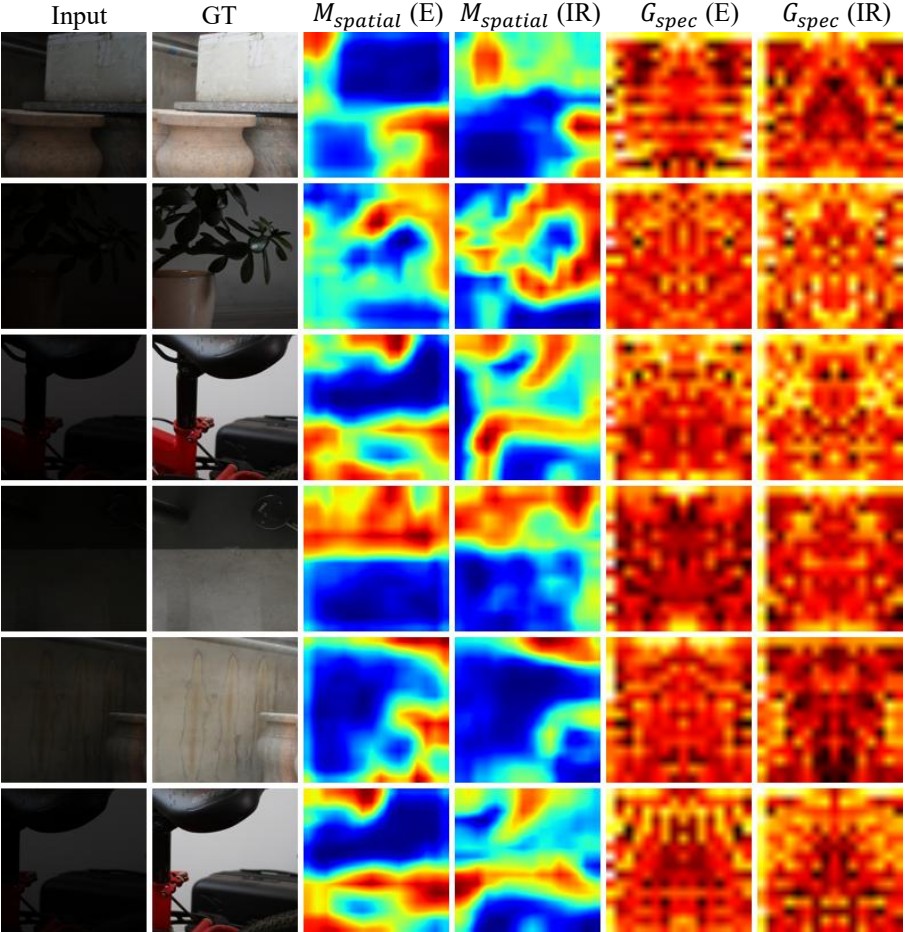

*Figure 7.* Visualization of spatial filtering and frequency-band gating in the S2DG Translator.

Fig. 8 shows that generated and real modality features form overlapping distributions in the latent space, indicating strong representation alignment.

Fig. 9 compares enhancement results using generated and real auxiliary modalities. The results are visually consistent in terms of brightness and structural details, suggesting that the generated modalities provide comparable guidance to the enhancement network.

## E. Additional Zero-Shot Qualitative Results

This section presents additional zero-shot qualitative comparisons to evaluate model generalization under unseen low-light conditions. All methods are directly applied without fine-tuning.

As shown in Fig. 10, our method produces more consistent enhancement results across datasets, with better recovery of fine textures and structural details. Compared with other methods, the outputs exhibit fewer artifacts and more stable brightness, indicating stronger robustness and cross-dataset generalization.

## F. Robustness to Missing Modalities

We analyze the impact of different auxiliary modalities during training, as well as the robustness of the proposed model under varying modality availability at inference time. Table 8 reports quantitative results on DID, SDSD-Indoor, and SDSD-Outdoor datasets under different combinations of training and inference modalities.

| IR | $Z^{ir}$ | $\hat{Z}^{ir}$ | Event | $Z^{evt}$ | $\hat{Z}^{evt}$ |

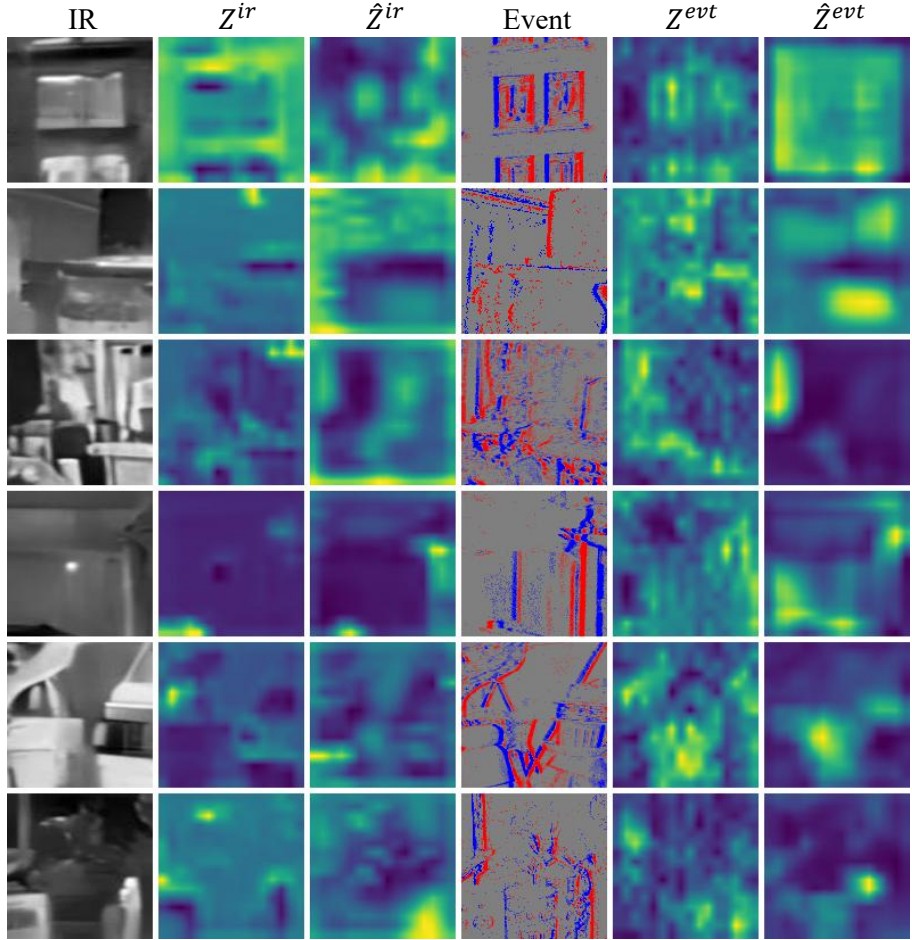

*Figure 8.* Feature distribution comparison between real and synthesized auxiliary modalities.

**Effect of Training Modalities.** We first evaluate how auxiliary modalities contribute during training when only RGB inputs are available at inference time (Table 8A). Compared with RGB-only training, introducing event or infrared supervision consistently improves performance across all datasets. In particular, infrared modality provides more noticeable gains than event data, especially on the DID dataset, suggesting its effectiveness in capturing low-light structural information. Training with all modalities achieves the best overall performance, demonstrating that multimodal supervision enables the model to learn more robust and informative representations even when auxiliary modalities are absent at test time.

**Inference Robustness under Missing Modalities.** We further investigate the robustness of the model under different inference-time modality configurations, with all modalities available during training (Table 8B). Notably, the model maintains strong performance under RGB-only inference, indicating that it does not rely on auxiliary modalities to function effectively. When event or infrared data are available at inference, the performance further improves, with full-modality inference achieving the best results. These observations validate that the proposed model supports flexible inference and remains robust under missing-modality scenarios, making it suitable for practical deployment where modality availability may vary.

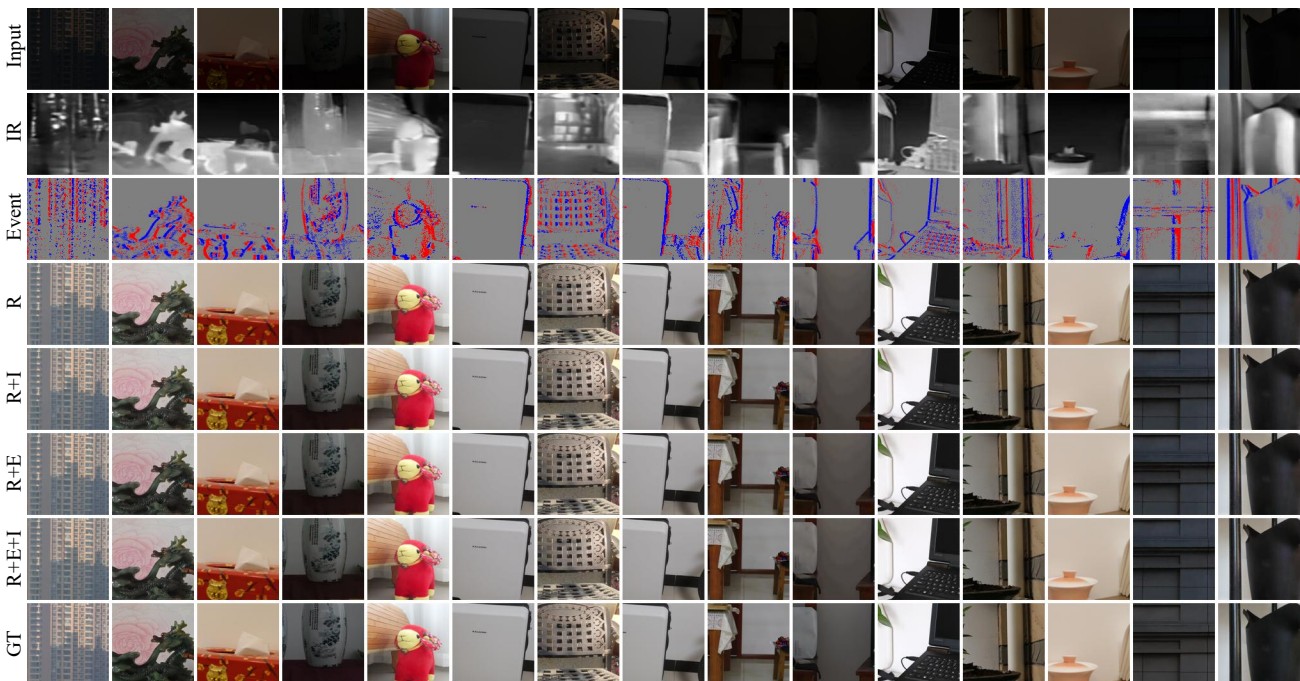

*Figure 9.* Qualitative comparison of enhancement results from AMNet using different auxiliary modalities.

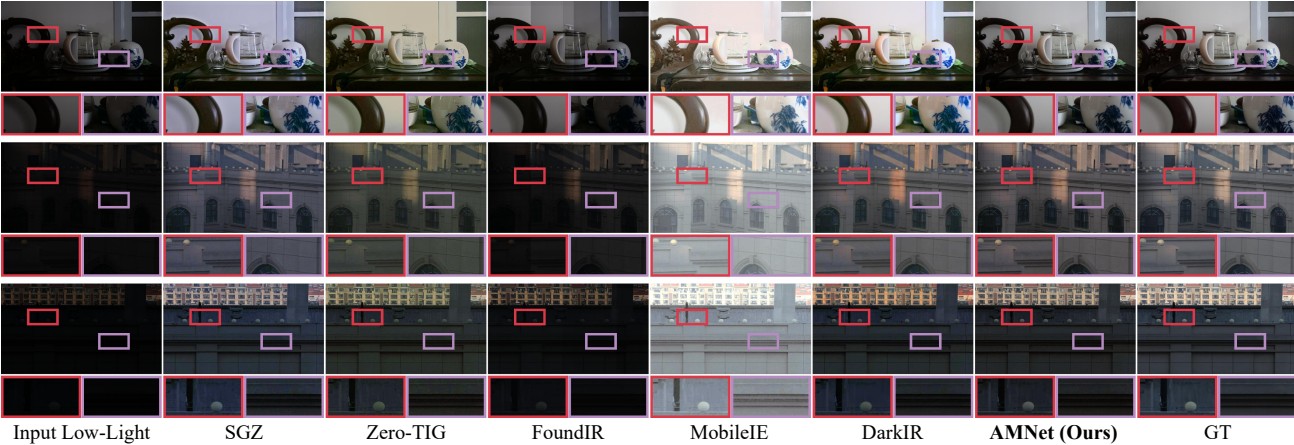

*Figure 10.* Qualitative comparison of zero-shot low-light video enhancement on unseen datasets. Without any finetuning, AMNet consistently recovers clearer structures and finer details than representative restoration foundation models, producing results closer to the ground truth.

*Table 8.* Ablation study on the effect of training modalities and inference robustness under different modality availability settings.

| Setting | Modalities | | | DID | | SDSD-Indoor | | SDSD-Outdoor | |
| --- | --- | --- | --- | --- | --- | --- | --- | --- | --- |
| | RGB | EVT | IR | PSNR | SSIM | PSNR | SSIM | PSNR | SSIM |
| **A. Effect of Training Modalities (RGB-only Inference)** | | | | | | | | | |
| RGB-only | ✓ | | | 29.78 | 0.941 | 27.96 | 0.912 | 26.07 | 0.824 |
| + Event | ✓ | ✓ | | 29.83 | 0.945 | 28.05 | 0.913 | 26.10 | 0.825 |
| + IR | ✓ | | ✓ | **30.12** | **0.946** | **28.32** | **0.915** | **26.18** | **0.829** |
| Full Modality | ✓ | ✓ | ✓ | **30.35** | **0.948** | **28.57** | **0.917** | **26.25** | **0.833** |
| **B. Inference Robustness (Full-Modality Training)** | | | | | | | | | |
| RGB-only | ✓ | | | 30.35 | 0.948 | 28.57 | 0.917 | 26.25 | 0.833 |
| + Event | ✓ | ✓ | | **31.25** | **0.955** | **28.75** | **0.918** | **26.35** | **0.836** |
| + IR | ✓ | | ✓ | 30.45 | 0.949 | 28.65 | 0.917 | 26.28 | 0.834 |
| Full Modality | ✓ | ✓ | ✓ | **31.29** | **0.955** | **28.82** | **0.921** | **26.29** | **0.842** |

