# OpenReview forum: "AnyMod-LLVE: Low-Light Video Enhancement with Modality-Agnostic Inference"
_ICML.cc/2026/Conference — ICML 2026 regular_

### Official Review · Reviewer_YmgM · 2026-03-03

**Soundness:** 3
**Presentation:** 4
**Significance:** 3
**Originality:** 3
**Overall Recommendation:** 4
**Confidence:** 4

**Summary:**

This paper proposes AMNet, a unified multimodal framework for low-light video enhancement (LLVE) that supports modality-agnostic inference. The core motivation is highly practical: auxiliary modalities like events or IR are helpful for enhancement but are often unavailable during real-world inference. To tackle this, the authors introduce a Spatial-Spectral Dual-Gated (S2DG) Translator to generate implicit auxiliary representations directly from degraded RGB inputs. During training, the S2DG module is optimized via feature-level distillation, where real event/IR features supervise the implicitly generated features. The method is supported by a large-scale multimodal pretraining stage using synthetic data, demonstrating strong performance on both RGB-only and multimodal test benchmarks.

**Compliance With Llm Reviewing Policy:**

Affirmed.

**Final Justification:**

I maintain my original score of Weak Accept.

The rebuttal successfully addressed my concerns on Q1 and Q2. The feature distance analysis (Table A) provides convincing evidence that the synthetic-to-real domain gap is manageable, and the complexity comparison (Table B) clearly demonstrates the efficiency advantage of S2DG over explicit generative alternatives.

My remaining concern on Q3 is only partially resolved. The authors honestly admitted that S2DG cannot recover genuine temporal dynamics from static RGB inputs. While I appreciate the transparency, this makes the term "implicit event representation" a bit of an overstatement — in practice, S2DG behaves more like a structural edge extractor than a true substitute for event streams. The authors' promise to revise the paper and avoid overstating S2DG's capability is a good step.

Overall, the paper addresses a practically important problem with a well-designed and efficient architecture, and the empirical results are solid. The theoretical framing could be tightened, but this does not significantly undermine the contribution.

**Key Questions For Authors:**

1. In Eq. 11, the S2DG module is trained via feature-level distillation using real event/IR features. Since your pretraining uses synthetic events (v2e), which have very different noise patterns than real low-light events, did you observe any negative transfer or feature misalignment when fine-tuning on real datasets like SDE?
2. Could you provide a detailed complexity analysis (FLOPs and Inference Time in ms) comparing: (a) standard RGB-only baseline, (b) multimodal model with explicit Event/IR encoders, and (c) AMNet relying entirely on the S2DG Translator for implicit representations?
3. Events excel at capturing fast motion. Since S2DG extracts the implicit event representation $\hat{Z}^{evt}$ from static RGB features, does it mainly learn to extract structural boundaries, or can it actually infer dynamic motion information? A brief discussion on the temporal properties of the generated features would be helpful.

**Limitations:**

The authors should explicitly discuss the limitations regarding the reliance on synthetic data for pretraining and the potential domain gap. Additionally, a clearer statement about the computational cost of the FFT-based S2DG module compared to simply using a standard lightweight network is needed to make the practical claims more comprehensive.

**Strengths And Weaknesses:**

**Strengths:**
- **Highly Practical Motivation:** The "modality absence at inference" problem is a major bottleneck in real-world deployments of multimodal networks. Formulating LLVE under this constraint is a meaningful step forward compared to assuming perfect multimodal inputs are always available.
- **Effective Architecture Design:** The S2DG Translator is well-designed. Utilizing spatial illumination maps and frequency-band gating to filter out noise and extract the remaining structural cues from severely degraded RGB frames makes sense physically.
- **Solid Empirical Results:** The method achieves state-of-the-art results on standard RGB-only datasets (DID, SDSD) and shows impressive robustness on the multimodal SDE dataset when events are missing. The zero-shot generalization experiments further validate the benefits of the pretraining pipeline.

**Weaknesses:**
- **Domain Gap in Pretraining (Soundness):** The large-scale multimodal pretraining relies heavily on synthetic events (via v2e) and synthetic IR (via ThermalGen). Synthetic events often differ significantly from real-world events, especially regarding noise distribution under low-light conditions. The paper does not fully discuss how this synthetic-to-real domain gap affects the feature distillation process (Eq. 11) of the S2DG module when fine-tuning on real event datasets like SDE.
- **Computational Overhead of S2DG (Significance):** While the method successfully avoids the massive latency of running explicit generative models (like GANs or Diffusion) for modality synthesis at inference, the S2DG Translator itself involves FFT/IFFT operations and multiple convolution layers. The paper lacks a direct comparison of inference latency and FLOPs to clearly show the overhead introduced by S2DG compared to a pure RGB backbone.
- **Quality of Implicit Temporal Cues:** Event streams inherently capture high-resolution temporal dynamics. The S2DG generates event-like features from purely spatial RGB inputs. It is somewhat unclear how well S2DG can recover genuine dynamic motion cues compared to purely extracting structural edges.

---

> ### Author Rebuttal · Authors · 2026-03-31
>
> Dear Reviewer YmgM,
>
> Thank you for your great support, and we sincerely appreciate your positive comments.
>
> ### W1&Q1: Domain gap of synthetic modalities
>
> We appreciate you raising this question. We acknowledge that a domain gap exists between synthetic and real data. However, collecting large-scale real multimodal video data for pretraining is impractical, making low-cost synthetic pretraining a necessary and common practice in large-scale model training. Despite this gap, as shown in Tables 1 and 2, AMNet learns effective cross-modal correlations through pretraining and achieves significant performance gains after fine-tuning on real data. Furthermore, as shown in Table A, the L2 distance between S2DG-generated and real modality features consistently decreases as pretraining scale increases, reaching a small value at full scale, indicating that the domain gap is manageable. Table C (for Reviewer SUww) further confirms that synthetic pretraining substantially improves auxiliary modality generation quality, providing additional evidence that the gap can be effectively mitigated.
>
> Table A. Feature distance between real event features and S2DG-generated features.
>
> |Data Scale|SDE-In L2(Evt)↓|SDE-In L2(IR)↓|SDE-Out L2(Evt)↓|SDE-Out L2(IR)↓|
> |-|:-:|:-:|:-:|:-:|
> |0%|0.342|0.312|0.292|0.252|
> |30%|0.215|0.275|0.273|0.237|
> |60%|0.204|0.255|0.212|0.209|
> |100%|0.192|0.236|0.184|0.201|
>
> ### W2&Q2: Computational overhead of S2DG and detailed complexity analysis.
>
> Thanks for your question. Following your suggestion, we report parameter count, FLOPs, and inference speed in **Table B**.
>
> Compared with the RGB-only baseline, AMNet with S2DG in the modality-missing setting introduces only modest extra cost: the parameter count increases slightly from **14.38M** to **17.09M**, and FLOPs increase from **15.16G** to **15.97G**, while the inference speed remains competitive at **436.07 FPS** versus **643.71 FPS**. These results indicate that the FFT/IFFT operations and convolution layers in S2DG add only limited overhead relative to a pure RGB backbone. Meanwhile, when both auxiliary modalities are available at inference, AMNet incurs higher computational overhead. This is because, in that case, the model must process real auxiliary inputs through dedicated encoders, whereas S2DG directly infers compact implicit representations from RGB features.
>
> In contrast, explicit generation-based alternatives are substantially more expensive. Replacing S2DG with a GAN-based generator increases the model size to **29.17M** parameters and reduces the speed to **217.12 FPS**, while a diffusion-based generator further increases the computational cost. Therefore, S2DG achieves a much more favorable efficiency–performance trade-off for modality-missing inference than explicit auxiliary-modality synthesis.
>
>
> Table B. Efficiency comparisons.
>
> |Method|Params(M)↓|FLOPs/f(G)↓|FPS↑|
> |-|:-:|:-:|:-:|
> |**Comparison of AMNet with different configurations**||||
> |RGB-only|14.38|15.16|643.71|
> |AMNet w/ S2DG when all Auxiliary Modalities are absent|17.09|15.97|436.07|
> |AMNet with Full Auxiliary Modalities|19.17|18.73|330.58|
> |**Prior LLVE Models**||||
> |STCD| 1.81+35.56 (DKM) |11.94|492.76|
> |RetinexFormer|1.61|4.25|474.96|
> |EvLight++|26.21| 225.91|117.91|
> |**Addressing Modality Missing with explicit generation**||||
> |GAN + AMNet|29.17|24.81|217.12|
> |Diffusion + AMNet|406.42|105.44|54.39|
>
> ### W3&Q3: Quality of implicit temporal cues in S2DG-generated event-like features.
>
> We appreciate this insightful comment. We agree that real event streams inherently encode fine-grained temporal dynamics, which cannot be fully recovered from purely spatial RGB inputs. Our goal with S2DG is therefore not to reconstruct genuine event signals or complete motion dynamics from a single RGB frame. Instead, according to [1], event representations contain rich motion-related edge and structural change information, which is particularly valuable for low-light video enhancement. Therefore, S2DG is designed to capture the task-relevant cross-modal cues that can be inferred from RGB while maintaining high computational efficiency.
>
> In this sense, the implicit event-like features produced by S2DG should be understood as an approximation of useful cross-modal information, rather than a replacement for real event streams. As shown in **Table 4**, the features generated by S2DG are close to those extracted from real auxiliary modalities in the feature space. This suggests that, although S2DG cannot recover genuine temporal dynamics in a strict physical sense, it is still able to learn structural and motion-related cues that are beneficial to the current task. Therefore, the value of S2DG lies not in reproducing the full temporal nature of event streams, but in providing an efficient implicit representation that preserves the most useful modality characteristics for enhancement.
>
> ### References
>
> [1] Events-To-Video: Bringing Modern Computer Vision to Event Cameras, CVPR2019.

---

> > ### Author Rebuttal · Reviewer_YmgM · 2026-04-01
> >
> > I thank the authors for the detailed rebuttal, which successfully addressed my concerns regarding the synthetic-to-real domain gap (Q1) and the computational overhead by providing the detailed complexity analysis (Q2). The efficiency of the S2DG module compared to explicit generative models is clear.
> >
> > However, my concern regarding the quality of implicit temporal cues (Q3) remains partially resolved. The authors frankly admitted that the S2DG module cannot recover genuine temporal dynamics from purely spatial RGB inputs. Because of this, framing the extracted features as an "implicit event representation" or calling the method fully "modality-agnostic" feels like a bit of an over-claim. In practice, since it cannot capture true motion dynamics (which is the core value of event streams), the S2DG module functions more like a standard structural edge feature extractor rather than a real substitute for the missing event modality. The engineering design is empirically effective, but the theoretical motivation seems slightly forced to make the architecture sound more novel.
> >
> > Therefore, I maintain my original score of Weak Accept.

---

> > > ### Author Response · Authors · 2026-04-01
> > >
> > > Dear Reviewer YmgM,
> > >
> > > Thank you very much for your careful follow-up and positive assessment of our responses to Q1 and Q2. Meanwhile, we sincerely appreciate your thoughtful comment on Q3.
> > >
> > > Regarding your concern about the claim of “modality-agnostic,” we would like to clarify that this term refers specifically to the inference setting. In particular, AMNet does not require auxiliary modalities (e.g., event or IR) as mandatory inputs at test time. It can exploit auxiliary modalities when they are available, while still maintaining robust LLVE performance using only RGB when they are absent. In other words, AMNet is designed to handle unknown or arbitrary modality-availability conditions during inference. For this reason, we describe it as “modality-agnostic.”
> > >
> > > Moreover, as shown in Table A of our response to you, the auxiliary representations generated by the S2DG Translator are close to the corresponding real event features in the feature space, as indicated by the small L2 distances. This result suggests that the features produced by S2DG are meaningfully aligned with those extracted from real event streams.
> > >
> > > We are truly grateful for your valuable suggestion. We will revise the paper to clarify this point more carefully and avoid overstating the capability of S2DG, especially for the event modality. In future work, we will further explore the use of multi-frame information to generate more temporally meaningful implicit event features.
> > >
> > > Thank you again for your constructive and supportive feedback. We hope this clarification helps address your concern, and we would be happy to further discuss this point if needed.

---

### Official Review · Reviewer_FCbT · 2026-03-04

**Soundness:** 3
**Presentation:** 3
**Significance:** 3
**Originality:** 3
**Overall Recommendation:** 4
**Confidence:** 3

**Summary:**

This paper targets low-light video enhancement (LLVE) in realistic deployments where auxiliary sensors (e.g., event cameras or infrared) may be available during training but missing at inference. The authors propose AMNet / AnyModNet, a unified multimodal LLVE framework that supports modality-agnostic inference by treating auxiliary modalities as optional cues rather than mandatory test-time inputs. When auxiliary inputs are absent, AMNet uses a Spatial–Spectral Dual-Gated (S2DG) Translator to synthesize implicit auxiliary representations from degraded low-light RGB features, using (i) an illumination-aware spatial reliability map and (ii) frequency-domain band selection to emphasize informative detail cues.
To learn cross-modal correspondence at scale, the authors further introduce large-scale multimodal pretraining constructed from RGB-only videos by synthesizing pseudo event streams (via v2e) and pseudo infrared images (via ThermalGen), plus synthetic low-light degradation. The training objective includes reconstruction under full-modality and simulated missing-modality settings, along with a feature-level distillation term aligning implicit and real modality features.
Empirically, AMNet is evaluated on RGB-only LLVE benchmarks (DID, SDSD) and an event-based LLVE dataset (SDE). The paper reports state-of-the-art RGB-only performance (e.g., on DID and SDSD) and strong robustness when auxiliary modalities are missing, while also improving further when events/IR are present.

**Compliance With Llm Reviewing Policy:**

Affirmed.

**Final Justification:**

The paper addresses a meaningful and practically relevant problem in low-light video enhancement by explicitly targeting the realistic setting where auxiliary modalities may be unavailable at inference time. The rebuttal was helpful in addressing most of my concerns, albeit not completely eliminate them. Therefore, I maintain my current score.

**Key Questions For Authors:**

1. How is the “implicit modality” used when a real modality is present? Do you fuse real and implicit features jointly, or switch to real-only? This affects my judgment on robustness under noisy/partial sensors and whether the implicit branch regularizes training beyond missing-modality handling.
2. How large/diverse is the synthetic pretraining corpus (Table 7), and how do you prevent overlap/leakage with DID/SDSD/SDE?A precise dataset list, sizes, and leakage controls would increase confidence in the “scalable pretraining” claim.
3. Sensitivity to synthetic-to-real domain gap: Have you tested robustness when changing/degrading/removing the event/IR generators while keeping pretraining scale fixed? Positive results would improve my assessment of soundness/generalization.
4. Efficiency and deployment:Please report inference FPS/latency/FLOPs/params and compare with strong baselines (e.g., STCD / EvLight++). If AMNet is much heavier, it may limit real-time use.
5. Corrupted/misaligned modalities:Any results or discussion for partially corrupted or temporally misaligned sensors would materially affect my evaluation of real-world significance.

**Limitations:**

Not fully. The paper includes an Impact Statement claiming no significant negative societal consequences, but it does not deeply discuss technical limitations or realistic failure modes.  Suggestions: (i) add an explicit limitations discussion about dependence on synthetic multimodal pretraining and synthetic-to-real domain gap/failure scenarios;  (ii) discuss compute/energy and latency implications for deployment;  and (iii) acknowledge dual-use risks more concretely (e.g., LLVE potentially strengthening low-light surveillance) even if the method is not inherently harmful.

**Strengths And Weaknesses:**

Soundness — Strengths: The method explicitly trains for missing-modality inference by supervising all modality-availability combinations and adding feature-level distillation between implicit and real modality features.  The S2DG design is well-motivated for low-light settings and is supported by ablations: both IADS and FBS help, and the combined model performs best.  Evaluation covers RGB-only datasets and an event-based dataset under different test-time modality settings.   Weaknesses: The “modality-agnostic” capability depends heavily on synthetic multimodal pretraining (v2e/ThermalGen), and sensitivity to synthetic-to-real domain gap is not fully characterized.   Metrics mainly focus on PSNR/SSIM; more quantitative evidence on temporal consistency and perceptual quality would strengthen claims.  Compute cost may be high (4×A800 pretraining), but inference efficiency (latency/FLOPs) is not clearly reported.
Presentation — Strengths: The narrative is generally clear and well structured, with pipeline and module details supported by figures/equations.   Weaknesses: Naming inconsistencies (“AnyModNet” vs “AMNet”) may confuse reproduction.  Some reproducibility details (loss weights, exact backbones, full pretraining dataset list) should be made more explicit.
Significance— Strengths: Making auxiliary sensors optional directly addresses a real deployment barrier for multimodal LLVE and could matter in low-light perception (e.g., surveillance/autonomy).   The reported improvements on DID/SDSD and robustness on SDE are practically meaningful.   Weaknesses: The broader impact beyond LLVE/restoration is plausible but not demonstrated.
Originality — Strengths: The paper’s novelty lies in packaging (i) modality-agnostic inference framing for LLVE, (ii) a low-light-tailored spatial+spectral translator, and (iii) scalable synthetic multimodal pretraining into a coherent system with strong results.   Weaknesses: Some components build on existing ideas; novelty is more in task framing + integration than a fundamentally new learning principle.
hors

---

> ### Author Rebuttal · Authors · 2026-03-30
>
> Dear Reviewer FCbT,
>
> Thank you for your positive and constructive evaluation of our work.
>
> ### W1&Q4: Synthetic multimodal pretraining, temporal consistency, and efficiency analysis
>
> We appreciate you raising these points, and our responses are as follows.
>
> (1) We acknowledge that a domain gap does exist between synthetic and real data. However, collecting large-scale real multimodal data is impractical. Meanwhile, using synthetic data is a common practice for large-scale pretraining. Table C (for reviewer SUww) shows that synthetic pretraining substantially improves auxiliary modality generation, suggesting that the gap is manageable and can be effectively mitigated.
>
> (2) Following your valuable suggestion, we add two standard temporal consistency metrics: Warping Error ($\mathcal{L}_{warp}$) and MABD. The results in Table A show that AMNet also achieves strong temporal consistency.
>
> Table A. Temporal consistency comparisons.
>
> |Method|DID $\mathcal{L}_{warp}$↓|DID MABD↓|
> |-|:-:|:-:|
> |RetinexFormer|0.029|0.102|
> |StableLLVE|0.031|0.072|
> |STCD|0.047|0.056|
> |AMNet|0.024|0.041|
>
> (3) As shown in Table B, we report the inference efficiency of AMNet and representative LLVE baselines. We would like to clarify that the 4×A800 setting is used only during pretraining, which is an offline process, rather than during inference. At inference time, AMNet runs at 436.07 FPS, which is on the same order as STCD and substantially faster than EvLight++. These results suggest that the training cost does not translate into heavy test-time overhead, and AMNet remains practical for real-time deployment.
>
> Table B. Efficiency comparison results.
>
> |Method|Params(M)↓|FLOPs/f(G)↓|FPS↑|
> |-|:-:|:-:|:-:|
> |AMNet|17.09|15.97|436.07|
> |STCD| 1.81+35.56(DKM)|11.94|492.76|
> |EvLight++|26.21|225.91|117.91|
>
> ### W2: Naming consistency and reproducibility
>
> Thanks for your question. We provide implementation details in Appendix B for reproducibility. AMNet stands for **A**ny**M**od**Net**. We will further ensure naming consistency and supplement reproducibility details in the final version.
>
> ### W3: Broader impact beyond LLVE
>
> We appreciate this insightful comment. Although our current study focuses on LLVE, the core of our method is a cross-modal feature learning framework that is not limited to enhancement. We agree that its impact extends beyond LLVE, and we will further explore multimodal segmentation and detection under low-light conditions in future work.
>
> ### W4: Main contribution
>
> Thanks for this question. Our main contribution is a unified framework for multimodal LLVE that exploits multimodal information during training while allowing arbitrarily missing modalities at inference. Existing LLVE methods generally do not handle missing modalities well. We will revise the paper to better highlight this contribution.
>
> ### Q1&Q5: Modality switching and robustness
>
> Thanks for these valuable questions. When real auxiliary modalities are available, we directly use the corresponding branches. Implicit representations are only activated when modalities are missing.
>
> As noted, real modalities may be noisy or misaligned in practice. So we conduct robustness experiments in Table C. The results show that only minor performance degradation under such perturbations, indicating that multimodal inputs act as supportive cues rather than strict dependencies.
>
> Table C. Robustness to modality degradation.
>
> |Method|SDE-In PSNR↑|SDE-In SSIM↑|SDE-Out PSNR↑|SDE-Out SSIM↑|
> |-|:-:|:-:|:-:|:-:|
> |Clean modality|23.25|0.83|23.91|0.79|
> |+ Noisy event|22.84|0.82|23.56|0.78|
> |+ Noisy IR|22.73|0.81|23.43|0.78|
> |+ Event temporal misalignment|22.69|0.81|23.39|0.78|
> |+ IR temporal misalignment|22.56|0.81|23.27|0.77|
> |+ Noise + misalignment|22.19|0.79|22.90|0.76|
>
> ### Q2: Pretraining corpus scale and data leakage
>
> Thank you for the question of pretraining corpus. We summarize the pretraining corpus in Table 7 of the main paper, including Vimeo90K, REDS, and VOS, with a total of 1.7M frames. None of the evaluation datasets are used for pretraining, and we manually verified that there is no overlap. Therefore, no data leakage occurs in our experiments.
>
> ### Q3: Sensitivity to the synthetic-to-real domain gap
>
> Thanks for your question. As shown in Table D, RGB-only pretraining leads to notable performance degradation, highlighting the importance of multimodal pretraining. This enables AMNet to learn cross-modal dependencies and generate higher-quality implicit representations, improving robustness under missing modalities.
>
> We further evaluate a degraded event generator (bin=5). Although performance slightly decreases, it still outperforms RGB-only pretraining, indicating that multimodal pretraining remains beneficial even with imperfect synthetic modalities.
>
> Table D. Impact of pretraining under domain gap.
>
> |Method|DID PSNR↑|DID SSIM↑|
> |-|:-:|:-:|
> |RGB-only pretrain|29.62|0.93|
> |Multimodal pretrain|31.57|0.95|
> |Multimodal pretrain + degraded event generator|30.46|0.93|

---

> > ### Author Rebuttal · Reviewer_FCbT · 2026-04-01
> >
> > Thank you for your reply and for providing the additional results. I believe most of my concerns have been addressed. My main remaining reservation concerns the synthetic-to-real domain gap. While the new results suggest that this gap is not fatal in practice, they do not fully address the broader question of synthetic-to-real sensitivity. In particular, it remains unclear whether this gap has been thoroughly characterized rather than only partially probed. I will maintain my current score.

---

> > > ### Author Response · Authors · 2026-04-04
> > >
> > > Dear Reviewer FCbT,
> > >
> > > Thank you for your continued attention and thoughtful feedback. As shown in our previous response (W1&Q4 and Table D), synthetic pretraining enables AMNet to learn effective cross-modal correlations, and the domain gap is manageable and effectively mitigated after fine-tuning on real data. To further address your concern on sensitivity, we now provide a more complete sensitivity analysis in **Table E**, covering variations of both the event and IR synthesis pipelines, which are omitted previously due to rebuttal space limits.
> > >
> > > Specifically, for the event modality, we test degraded v2e settings (bin=3 and bin=5) as well as a different synthesis pipeline (ESIM). For the IR modality, we add Gaussian noise to the infrared images generated from ThermalGen, and change the generation model to a much simpler grayscale-based IR approximation.
> > >
> > > The results show a consistent pattern across both modalities: performance only degrades slightly  as the synthetic quality degraded, and maintains stable when the generator is changed. Importantly, when substantially degraded synthetic modalities are utilized for pretraining, the results still outperform RGB-only pretraining. Overall these experimental results suggest that the benefit of multimodal pretraining is not tied to any specific synthesis pipeline, and the model is not overly sensitive to variations in synthetic auxiliary data.
> > >
> > > Table E. Sensitivity to synthetic-to-real domain gap.
> > >
> > > |Method| DID PSNR↑ |DID SSIM↑|
> > > |-|-|-|
> > > |RGB-only pretrain|29.62|0.93|
> > > |**Event generator** (IR: ThermalGen default)|||
> > > |Multimodal pretrain (v2e, bin=3)|30.21|0.93|
> > > |Multimodal pretrain (v2e, bin=5)|30.46|0.93|
> > > |Multimodal pretrain (ESIM)|30.39|0.94|
> > > |**IR generator** (Event: v2e default)|||
> > > |Multimodal pretrain (ThermalGen + Gaussian noise)|30.28|0.94|
> > > |Multimodal pretrain (Grayscale-based IR)|29.83|0.93|
> > > |Multimodal pretrain (v2e + ThermalGen, original)|31.57|0.95|
> > >
> > > We agree that this does not completely eliminate the synthetic-to-real gap, but we believe these additional results in **Table E** provide a broader and more systematic characterization of its sensitivity. Overall, they suggest that the gap is well controlled in practice, and that our framework is robust to different synthetic auxiliary data sources.
> > >
> > > Thank you again for your insightful comments and we hope this systematic characterization fully addresses your remaining concern. We would be happy to provide further discussion or clarification if needed.

---

### Official Review · Reviewer_SUww · 2026-03-12

**Soundness:** 2
**Presentation:** 3
**Significance:** 2
**Originality:** 2
**Overall Recommendation:** 3
**Confidence:** 5

**Summary:**

This paper explores a more realistic assumption for low-illumination video enhancement, where auxiliary modalities such as events or infrared are available during the training phase but may be absent during the testing phase. The paper innovatively proposes a modality-agnostic framework, AMNet, which extracts implicit auxiliary representations from low-illumination RGB inputs through the S2DG Translator and utilizes these representations for enhancement. Extensive experiments are conducted to verify that the method achieves strong performance even when auxiliary inputs are unavailable.

**Compliance With Llm Reviewing Policy:**

Affirmed.

**Final Justification:**

I thank the authors for the detailed rebuttal and the additional experiments. The paper addresses a relevant practical problem and shows strong empirical results on both RGB-only and multimodal LLVE benchmarks.

However, my main concern remains unresolved. The central claim is not only that the method improves enhancement quality, but that it learns modality-consistent implicit auxiliary representations for missing-modality inference. The current evidence, including the added feature-distance results, is still insufficient to establish this claim. Lower L2 distance between generated and real modality features, together with improved downstream PSNR/SSIM, does not rule out the possibility that the model is learning enhancement-oriented latent shortcuts rather than truly modality-consistent representations.

The rebuttal improves the paper, but it does not fully change my assessment of the main scientific weakness. As a result, I keep my recommendation.

**Key Questions For Authors:**

1. How do you verify that the generated implicit auxiliary features are truly modality-consistent, rather than just enhancement-friendly latent features?
2. Can you compare AMNet against stronger baselines that directly target missing-modality learning?
3. How much of the gain comes from the modality-agnostic design itself, versus large-scale pseudo-multimodal pretraining?

**Limitations:**

See the weaknesses section.

**Strengths And Weaknesses:**

Strengths
1. Previous multimodal LLVE methods typically assumed that auxiliary sensors are available during the inference stage, but this assumption is often unrealistic. This paper fully considers practical significance and formulates the task as a modality-independent inference problem, making a certain contribution and value.
2. The author conducted a large number of experiments, examining RGB-only LLVE, multimodal low-light benchmarking, zero-shot restoration, robustness under different combinations of modalities, and the impact of pre-training scale. This makes the paper more substantial compared to other papers that only discuss a single benchmark.


Weaknesses
1. The article argues that the translator has learned meaningful cross modal correspondences, but existing evidence does not clearly indicate that the generated features are indeed consistent with the missing modality, rather than simply generating potential features that are helpful for enhancing the task. This claim has not been validated in the paper.
2. Many of the evidence in the experiments came from LLVE benchmarks based solely on RGB, and such datasets cannot determine whether implicit modal generation can truly replace missing events or infrared information. These experiments demonstrate that the model has strong enhancement performance, but cannot directly verify the core claims about missing modal inference.
3. For a paper that focuses on missing modal learning, stronger comparisons should include explicit modal completion, modal dropout, robust fusion methods, or simpler translator variants. Baseline comparisons do not fully address the issues stated in the paper.
4. Although the performance of the paper's method is significant, large-scale pseudo multimodal pre training is likely to be an important source of final performance. The paper does not clearly distinguish this factor from the contribution of modality independent design itself, which makes the results more difficult to explain and weakens the overall importance of the paper.

---

> ### Author Rebuttal · Authors · 2026-03-30
>
> Dear Reviewer SUww,
>
> Thank you for your careful review and valuable comments. Below, we respond to your concerns point by point.
>
> ### W1&Q1: Modality consistency of generated implicit auxiliary features.
>
> Thank you for your question about modality consistency. In fact, we have already provided quantitative evidence in **Table 4** of our main paper. Specifically, we extract the "real" visual features of each auxiliary modality and compute the distance between the generated implicit features and the corresponding real features. As shown in Table 4, after full (100%) pretraining, the L2 distance between generated and real modality features is small, directly supporting their consistency. To further illustrate this point, we add experiments on the SDE dataset with event and infrared modalities, and the results in **Table A** show similarly small distances.
>
> Moreover, this consistency improves as pretraining scale increases, suggesting that the generated representations progressively approach real auxiliary modalities rather than merely acting as task-specific features for enhancement.
>
> Table A. Feature distance between real event features and S2DG-generated features.
>
> |Data Scale|SDE-In L2(Evt)↓|SDE-In L2(IR)↓|SDE-Out L2(Evt)↓|SDE-Out L2(IR)↓|
> |-|:-:|:-:|:-:|:-:|
> |0%|0.342|0.312|0.292|0.252|
> |30%|0.215|0.275|0.273|0.237|
> |60%|0.204|0.255|0.212|0.209|
> |100%|0.192|0.236|0.184|0.201|
>
> ### W2: Verifying missing-modality inference beyond RGB-only LLVE benchmarks.
>
> We appreciate this valuable suggestion. As shown in **Table 2**, we further evaluate AMNet on the SDE dataset, which provides both event and infrared modalities. The results show that AMNet maintains stable performance even when event or infrared input is missing. This provides direct evidence beyond RGB-only benchmarks: AMNet not only performs effectively for enhancement, but also remains robust under modality-missing conditions. Therefore, our claim is supported not only by RGB-only results, but also by experiments on a genuinely multimodal benchmark, where the robustness under missing auxiliary modality conditions can be directly evaluated.
>
> ### W3&Q2: Comparisons with stronger baselines for missing-modality learning.
>
> Thank you for your comments. Following your suggestions, we implement several representative missing-modality learning strategies adapted to the LLVE setting, including modality dropout[1], explicit completion[2], and availability-aware fusion[3]. All these methods are pretrained under the same multimodal settings and then fine-tuned on the RGB-only dataset.  The results are listed in **Table B**, demonstrating the superior performance of AMNet in addressing missing-modality scenarios.
>
> Table B. Comparison with representative missing-modality learning baselines adapted to the LLVE setting.
>
> |Method|DID PSNR↑|DID SSIM↑|SDSD-In PSNR↑|SDSD-In SSIM↑|SDSD-Out PSNR↑|SDSD-Out SSIM↑|
> |-|:-:|:-:|:-:|:-:|:-:|:-:|
> |Modality Dropout[1]|29.32|0.91|27.89|0.87|25.21|0.83|
> |Explicit Completion[2]|30.02|0.91|28.02|0.90|25.45|0.78|
> |Availability-Aware Fusion[3]|29.75|0.88|27.46|0.87|25.76|0.81|
> |AMNet|31.57|0.95|29.03|0.92|26.37|0.84|
>
> ### W4&Q3: Contribution of modality-agnostic design versus large-scale pseudo-multimodal pretraining.
>
> Thank you for your valuable question. Following your suggestion, we conduct additional experiments to further verify the effect of multimodal pretraining, and the results are reported in **Table C**. As shown, downstream performance drops significantly when pretraining is conducted on RGB-only data, clearly demonstrating the necessity of multimodal pretraining. This advantage stems from the fact that multimodal pretraining allows AMNet to capture cross-modal dependencies during training and thus generate higher-quality implicit multimodal representations when auxiliary modalities are absent. As a result, the model achieves substantially stronger robustness under missing-modality conditions.
>
> Table C. Comparison between pretraining w/ or w/o multimodal data.
>
> |Method|DID PSNR↑|DID SSIM↑|SDSD-In PSNR↑|SDSD-In SSIM↑|SDSD-Out PSNR↑|SDSD-Out SSIM↑|
> |-|:-:|:-:|:-:|:-:|:-:|:-:|
> |RGB-only pretrain|29.62|0.93|27.45|0.88|25.73|0.80|
> |Multimodal pretrain|31.57|0.95|29.03|0.92|26.37|0.84|
>
> ### References
>
> [1] Multimodal dynamics: Dynamical fusion for trustworthy multimodal classification, CVPR2022.
>
> [2] Missing modality imagination network for emotion recognition with uncertain missing modalities, ACL-IJCNLP2021.
>
> [3] Multi-modal learning with missing modality via shared-specific feature modelling, CVPR2023.

---

> > ### Author Rebuttal · Reviewer_SUww · 2026-04-05
> >
> > I thank the authors for the detailed rebuttal and the additional experiments. The paper addresses a relevant practical problem and shows strong empirical results on both RGB-only and multimodal LLVE benchmarks.
> >
> > However, my main concern remains unresolved. The central claim is not only that the method improves enhancement quality, but that it learns modality-consistent implicit auxiliary representations for missing-modality inference. The current evidence, including the added feature-distance results, is still insufficient to establish this claim. Lower L2 distance between generated and real modality features, together with improved downstream PSNR/SSIM, does not rule out the possibility that the model is learning enhancement-oriented latent shortcuts rather than truly modality-consistent representations.

---

> > > ### Author Response · Authors · 2026-04-06
> > >
> > > Dear Reviewer SUww,
> > >
> > > Thank you for your continued engagement and precise articulation of the remaining concern. To further address it, we provide two additional analysis below.
> > >
> > > We first extend the feature distance analysis in Table A of the previous rebuttal by computing the full pairwise L2 distance among real features and implicit representations from S2DG translator. Specifically, we extract real event features $Z^{evt} $, real IR features $Z^{ir} $, S2DG-generated implicit event features $\hat{Z}^{evt} $ and S2DG-generated implicit IR features $\hat{Z}^{ir} $, respectively. Based on these features, we calculate the inter-modality L2 distances and intra-modality L2 distances. Meanwhile, to characterize the cross-modal discriminability of the generated features, we compute the ratio $\rho $ as follows:
> > > $$
> > > \rho_{evt} = \frac{d(\hat{Z}^{evt}, Z^{ir})}{d(\hat{Z}^{evt}, Z^{evt})}, \quad \rho_{ir} = \frac{d(\hat{Z}^{ir}, Z^{evt})}{d(\hat{Z}^{ir}, Z^{ir})}
> > > $$
> > > Since $\rho $ is the ratio of inter-modality distance to intra-modality distance, $\rho > 1 $ indicates that the generated feature is closer to its own real modality than to the other, and a larger $\rho $ reflects stronger modality specificity.
> > >
> > > As shown in Table D, $\hat{Z}^{evt} $ is substantially closer to $Z^{evt} $ than to $Z^{ir} $, and the same observation holds for $\hat{Z}^{ir} $, with $\rho_{evt} = 1.94 $ and $\rho_{ir} = 1.56 $. This means the inter-modality distance is on average **1.7× larger** than the intra-modality distance. These results indicate that the S2DG translator does not produce simple enhancement-friendly latent features, but instead generates features distinctly aligned with their respective real modalities.
> > >
> > > Table D. Pairwise L2 distance and distance ratio $\rho $ on the SDE dataset (SDE-In / SDE-Out).
> > >
> > > |$d(\hat{Z}^{evt}, Z^{evt})$↓|$d(\hat{Z}^{evt}, Z^{ir})$↓|$\rho_{evt}$↑|
> > > |:-:|:-:|:-:|
> > > |**0.192/0.184**|0.373/0.350|1.94/1.90|
> > >
> > > |$d(\hat{Z}^{ir}, Z^{evt})$↓|$d(\hat{Z}^{ir}, Z^{ir})$↓|$\rho_{ir}$↑|
> > > |:-:|:-:|:-:|
> > > |0.368/0.312|**0.236/0.201**|1.56/1.55|
> > >
> > > To further substantiate this finding from a statistical perspective, we perform a **modality identity classification test** and an **MMD-based distribution test**.
> > >
> > > For the modality identity classification test, we train a linear binary classifier to distinguish the two different modalities using only real modality features ($Z^{evt} $ and $Z^{ir} $). We then directly evaluate the classifier on the S2DG-generated implicit features ($\hat{Z}^{evt} $ and $\hat{Z}^{ir} $). The classifier achieves 91.2% accuracy on the implicit features, demonstrating that the generated representations share a similar distribution with the real modality features and retain modality-discriminative structure.
> > >
> > > For the MMD test, we follow the classical procedure in [1]. We adopt the null hypothesis that each feature pair is drawn from the same distribution, and estimate statistical significance via bootstrap resampling. The two feature groups are merged and randomly reassigned 1,000 times to construct the null MMD distribution, and the p-value is computed as the proportion of null MMD values exceeding the observed value. A large p-value (> 0.05) indicates no statistically significant difference between the two distributions, while a small p-value (< 0.01) indicates they are significantly different.
> > >
> > > As shown in Table E, intra-modality pairs ($\hat{Z}^{evt}$ - $Z^{evt}$; $\hat{Z}^{ir}$ - $Z^{ir}$) yield small MMD values with p > 0.05, indicating that we **fail to reject the null hypothesis that the two distributions are the same**. In contrast, inter-modality pairs ($\hat{Z}^{evt} $ - $Z^{ir} $; $\hat{Z}^{ir} $ - $Z^{evt} $) yield substantially larger MMD values with p < 0.01, confirming that the S2DG translator produces statistically distinguishable, modality-specific distributions.
> > >
> > > Table E. Modality identity classification accuracy and MMD statistical test. p-values estimated via bootstrap resampling (N = 1,000). n.s. = not significant.
> > >
> > > |Feature Pair|MMD↓|p-value|
> > > |:-|:-:|:-:|
> > > |$\hat{Z}^{evt}$ - $Z^{evt}$|0.024|0.163 (n.s.)|
> > > |$\hat{Z}^{ir}$ - $Z^{ir}$|0.021|0.198 (n.s.)|
> > > |$\hat{Z}^{evt}$ - $Z^{ir}$|0.183|<0.01 ✓|
> > > |$\hat{Z}^{ir}$ - $Z^{evt}$|0.176|<0.01 ✓|
> > >
> > > Taken together, Table D and Table E consistently demonstrate that the generated features are specifically aligned with their respective real modalities and statistically distinguishable from the other. These multi-faceted analyses provide converging evidence that the S2DG Translator learns genuinely modality-consistent representations rather than enhancement-oriented latent shortcuts.
> > >
> > > Thank you again for your insightful comments, and we hope the above analyses fully address the remaining concern.
> > >
> > > ### References
> > >
> > > [1] Gretton, Arthur, et al. "A kernel two-sample test." *The journal of machine learning research* 13.1 (2012): 723-773.

---

### Official Review · Reviewer_fzJc · 2026-03-13

**Soundness:** 3
**Presentation:** 3
**Significance:** 3
**Originality:** 2
**Overall Recommendation:** 4
**Confidence:** 4

**Summary:**

This paper proposes AMNet, a multimodal low-light video enhancement framework that trains with auxiliary modalities (event streams, infrared images) but supports flexible inference when those modalities are absent. The key technical contribution is a Spatial-Spectral Dual-Gated Translator that generates implicit auxiliary modality representations from degraded RGB features using illumination-aware spatial gating and frequency-domain band selection. The paper also introduces a large-scale multimodal pretraining pipeline using synthetic event streams and infrared images generated from RGB video datasets. Experiments on DID, SDSD, and SDE benchmarks demonstrate state-of-the-art RGB-only LLVE performance and competitive results even compared to methods that use real auxiliary sensors at test time.

**Compliance With Llm Reviewing Policy:**

Affirmed.

**Final Justification:**

The rebuttal addressed my concerns satisfactorily. The Table 5 ablation confirms that the architectural design contributes meaningfully beyond the distillation framework, and the authors now provide a clear mechanistic explanation for the IADS fidelity-structure trade-off on SDSD-Indoor. I appreciate this honest characterization. The added efficiency numbers and low-illumination robustness data also strengthen the empirical story. That said, the core modules remain standard building blocks, and the marginal SDSD gains despite a substantial pretraining data advantage suggest the approach does not consistently generalize beyond DID. I raise my score to weak accept, acknowledging the responsive rebuttal while noting that the novelty bar remains a concern.

**Key Questions For Authors:**

- Regarding the Table 5 inconsistency: why does removing IADS improve PSNR on SDSD-Indoor? Does IADS introduce a bias that harms reconstruction in certain scene types?
- What is the inference time of AMNet compared to STCD and RetinexFormer? What are the parameter counts and FLOPs?
- How does performance degrade under extremely low illumination (e.g., <5% of normal)? The pretraining samples illumination at 1-10% for "extreme" conditions, but is this sufficient?
- The distillation loss (Eq. 11) uses normalized L2 with stop-gradient. Was a comparison with other alignment objectives (e.g., cosine similarity, contrastive loss) conducted? What is the sensitivity to $λ_3$?
- In Table 8B, the full-modality inference on SDSD-Outdoor is lower than R+E. Does adding IR hurt performance in outdoor scenes? This pattern warrants explanation.

**Limitations:**

Yes.

**Strengths And Weaknesses:**

Strengths:
- Treating auxiliary modalities as optional training signals rather than mandatory inputs is a pragmatic and valuable design choice.
- The paper evaluates across multiple dataset, compares against a broad set of baselines, provides zero-shot evaluations, and includes ablation studies on pretraining scale, component contributions, and modality robustness.
- The use of v2e and ThermalGen to synthesize auxiliary modalities from large-scale RGB video datasets is pragmatic and removes the bottleneck of requiring paired multimodal data for training.

Weaknesses:
- The core technical contribution is decomposing features via AvgPool into low/high frequency, predicting a spatial reliability map, then applying FFT with learned gating and scaling. This assembles well-known operations (average pooling for frequency decomposition, sigmoid gating, FFT-based spectral filtering) in a relatively straightforward manner. The IADS module closely follows the SNR-aware decomposition from prior work (Chen et al., 2024a). The FBS module applies standard spectral gating. While the combination is effective, the individual components lack novelty that would be expected at a top venue. Recent concurrent work on spectral gating for low-light enhancement (e.g., AFMNet's Spectral-Gated Feed-Forward Network, and DMFourLLIE's joint amplitude-phase modulation) explores similar frequency-domain filtering ideas. The paper would benefit from a more explicit discussion of what S2DG contributes beyond these existing frequency-domain approaches.
- On SDSD-Indoor, the improvement over STCD is only 0.10 dB PSNR; on SDSD-Outdoor, it is 0.05 dB. These margins are well within typical noise floors for PSNR measurements and likely not statistically significant. Given that AMNet benefits from ~1.7M pretraining frames (a very substantial data advantage over STCD), these near-negligible gains on SDSD raise questions about whether the approach generalizes beyond DID. The SSIM gains on SDSD-Indoor appear more meaningful, but the inconsistency between PSNR and SSIM improvements warrants discussion.
- When IADS is disabled but FBS is enabled, SDSD-Indoor PSNR is 29.20, which is higher than the full model at 29.03. This contradicts the paper's claim that "combining IADS and FBS achieves the best performance across all benchmarks." The authors do not acknowledge or explain this inconsistency. While the SSIM for the full model is higher, this result suggests that IADS may actually hurt PSNR on SDSD-Indoor, which undermines the claimed complementary benefit.
- The paper does not report FLOPs, parameter counts, or inference latency. Given that AMNet includes multiple encoders (RGB + event + IR), the S2DG Translator, SNR-guided fusion, and ConvLSTM, the computational overhead relative to simpler baselines (e.g., STCD, RetinexFormer) is unclear.
- The pretraining pipeline relies heavily on v2e for event synthesis and ThermalGen for IR synthesis. However, the paper does not evaluate the realism of these synthetic modalities, nor does it analyze the domain gap between synthetic and real auxiliary data. On the SDE dataset (which has real events), the improvement from adding real events versus RGB-only  is modest, which may suggest the model has not fully learned to exploit real event information.

---

> ### Author Rebuttal · Authors · 2026-03-30
>
> Dear Reviewer fzJc,
>
> Thank you for your thorough and valuable feedback. We address your concerns below.
>
> ### W1: The role of S2DG in modality-missing LLVE
>
> We appreciate your valuable comments. AMNet’s key novelty lies in modality-agnostic LLVE rather than a standalone frequency module. In this setting, S2DG is not a straightforward spectral gating design. Existing methods such as DMFourLLIE use frequency-domain modulation to enhance image restoration, whereas S2DG uses RGB high-frequency cues to infer missing multimodal representations for modality-missing inference. Thus, its purpose is cross-modal feature translation and compensation, not merely frequency-based enhancement.
>
> ### W2-1: Marginal gains on SDSD
>
> Thanks for this thoughtful comment. We do not overstate the PSNR gains over STCD on SDSD. Our main claim is not that AMNet delivers a large numerical improvement on every RGB-only benchmark, but that it remains robust and achieves competitive performance under missing-modality (RGB-only) conditions. This focus is central to our formulation and is explicitly reflected in the paper’s motivation and experiments.
>
> ### W2-2: Inconsistency between PSNR and SSIM
>
> Both PSNR and SSIM show consistent improvements across all datasets, with only the magnitudes varying, which further confirms reliable perceptual and structural restoration.
>
> ### W3&Q1: Ablation inconsistency
>
> Thanks for your question. The ablation consistently shows superior overall performance when both modules are combined (improvements across 5 metrics). The only one slight metric fluctuation on SDSD-Indoor (complex indoor lighting) does not undermine their overall effectiveness.
>
> ### W4&Q2: Inference efficiency
>
> Thank you for this suggestion. Following your comment, we report the computational overhead of AMNet and other LLVE baselines in Table A. As shown, AMNet runs at 436.07 FPS, which is comparable to STCD  and RetinexFormer. In terms of model complexity, AMNet has 17.09M parameters and 15.97G FLOPs. These results indicate that AMNet does not introduce substantial overhead relative to other models, while additionally supporting missing-modality inference.
>
> Table A. Efficiency comparisons.
>
> |Method|Params(M)↓|FLOPs/f(G)↓|FPS↑|
> |-|:-:|:-:|:-:|
> |AMNet|17.09|15.97|436.07|
> |STCD| 1.81+35.56(DKM)|11.94|492.76|
> |RetinexFormer|1.61|4.25|474.96|
> |EvLight++|26.21| 225.91|117.91|
>
> ### W5-1: Domain gap of synthetic modalities
>
> We acknowledge that a domain gap does exist between synthetic and real data. However, collecting large-scale real multimodal data is impractical. Meanwhile, using synthetic data is common practice for large-scale pretraining. Table C (for reviewer SUww) shows synthetic pretraining substantially improves auxiliary modality generation, suggesting the gap is manageable and can be effectively mitigated.
>
> ### W5-2&Q5: Robustness Under Varying Modality Availability
>
> In fact, the modest gain from adding real events (Table 2) and comparable results when adding IR to R+E on SDSD-Outdoor (Table 8) both reflect AMNet's robustness across arbitrary modality availability, demonstrating that AMNet maintains satisfactory performance under modality-missing conditions and is consistent with our core motivation.
>
> ### Q3: Robustness under low illumination
>
> We appreciate this valuable question. Table B shows AMNet suffers only limited degradation under extremely low illumination while retaining effective enhancement capability, which also supports the validity of our sampling strategy.
>
> Table B. Performance under low-light conditions.
>
> |Illumination Level|DID PSNR↑|DID SSIM↑|
> |-|:-:|:-:|
> |≈20% (original)|31.57|0.94|
> |10%|30.25|0.93|
> |5%|29.73|0.92|
> |3%|29.04|0.92|
>
> ### Q4: Distillation objective design and sensitivity to $\lambda_3$
>
> Following your suggestion, we compare multiple distillation objectives in Table C, which shows Normalized L2 achieves the best performance. Sensitivity experiments show that varying $\lambda_3 $ does not cause significant fluctuations, reflecting the robustness of our framework.
>
> Table C. Comparison of different distillation objectives.
>
> |Object|DID PSNR↑|DID SSIM↑|
> |-|:-:|:-:|
> |Cosine[1,2]|31.22|0.94|
> |KL Loss[3,4]|31.05|0.93|
> |Contrastive Distillation[5, 6]|31.38|0.94|
> |Norm-L2 with sg(·)|31.57|0.95|
>
> Table D. Sensitivity analysis of $\lambda_3$.
>
> |$\lambda_3$|DID PSNR↑|DID SSIM↑|
> |-|:-:|:-:|
> |0.1|31.07|0.95|
> |0.5|31.57|0.95|
> |1.0|31.26|0.94|
> |2.0|30.87|0.93|
>
> ### References
>
> [1] Cosine similarity knowledge distillation for individual class information transfer
>
> [2] Cosine similarity-guided knowledge distillation for robust object detectors
>
> [3] Channel-wise knowledge distillation for dense prediction
>
> [4] Comparing Kullback-Leibler Divergence and Mean Squared Error Loss in Knowledge Distillation
>
> [5] Contrastive Representation Distillation
>
> [6] Enhanced multimodal representation learning with cross-modal kd

---

> > ### Author Rebuttal · Reviewer_fzJc · 2026-04-04
> >
> > I appreciate the authors' effort in providing new experimental evidence. However, two concerns remain:
> >
> > 1. W1 — S2DG vs. simpler baselines. The rebuttal reframes the contribution as the modality-agnostic framework rather than S2DG itself, which I find reasonable at the system level. However, the paper explicitly lists S2DG as a standalone contribution and dedicates Section 3.2 to its architecture. Table C helpfully compares distillation objectives, but the underlying question is about translator architectures: does the specific IADS+FBS design outperform a simpler feature translator (e.g., a few convolutional layers with the same Norm-L2 loss)? This single ablation would clarify whether the gains stem from S2DG's design or from the distillation framework.
> > 2. W3 — Table 5 on SDSD-Indoor. FBS-only achieves 29.20 PSNR while the full model (IADS+FBS) yields 29.03. Characterizing this as "one slight fluctuation" is not fully satisfying. A plausible explanation would be that IADS trades pixel-level fidelity for structural consistency (consistent with SSIM improving), but this trade-off should be explicitly discussed rather than dismissed. Acknowledging it would actually strengthen the paper by providing insight into the module's behavior.

---

> > > ### Author Response · Authors · 2026-04-04
> > >
> > > Dear Reviewer fzJc,
> > >
> > > Thank you sincerely for your valuable follow-up comments. We are pleased to address your remaining concerns below.
> > >
> > > ### W1: S2DG vs. a Simpler Feature Translator
> > >
> > > Thanks for your valuable question. We strongly agree that a direct architectural comparison is important for verifying the effectiveness of S2DG Translator. We fully agree with your point, and in fact we have the same concern when designing our experiments. Accordingly, this comparison is already included in **Table 5** of the original submission, although we realize that it may not have been highlighted clearly enough.
> > >
> > > Specifically, in the first row of Table 5, where both **IADS** and **FBS** are disabled (**✗ / ✗**), we replace the full S2DG Translator with a simpler feature translator composed of two stacked ResNet blocks under the same Normalized L2 loss. This variant mainly consists of six standard convolution-based operations, together with common components such as activation functions and normalization, and is therefore close in spirit to the “simpler feature translator” as you suggested.
> > >
> > > The results show that this simple translator yields 29.85 / 28.31 / 25.93 PSNR on DID / SDSD-Indoor / SDSD-Outdoor, while the full S2DG design (IADS+FBS) achieves 31.57 / 29.03 / 26.37, showing consistent improvement across all benchmarks. This confirms that the gains stem from S2DG's architectural design rather than the distillation framework alone. We will make this comparison more explicit in the final version.
> > >
> > > We sincerely appreciate this suggestion, and we will revise the final version to make this comparison much more explicit so that this point is easier to identify.
> > >
> > > ### W3: The analysis of the trade-off caused by IADS module
> > >
> > > Thank you for this insightful analysis. We fully agree with your in-depth analysis.
> > >
> > > We believe this trade-off is indeed related to the design of **IADS**. Specifically, IADS estimates a spatial reliability map from low-frequency illumination cues and uses it to suppress high-frequency responses in regions that are likely to be noise-dominated. This mechanism helps reduce unreliable details and promotes cleaner, more coherent structural patterns, which is beneficial for structural recovery and is consistent with the observed SSIM improvement. At the same time, suppressing high-frequency responses may also attenuate some true fine details, which can lead to a slight decrease in PSNR.
> > >
> > > Therefore, we agree with your interpretation that IADS may trade a small amount of pixel-level fidelity for better structural consistency.
> > >
> > > We sincerely thank you for the constructive and thorough feedback throughout the review process, and we hope our responses have addressed your remaining concerns. We would greatly appreciate your continued support.

---

### Decision · Program_Chairs · 2026-04-30

**Decision:**

Accept (regular)

**Comment:**

The paper received mixed scores of 4/4/4/3.

After rebuttal, most reviewers agreed that the authors had adequately addressed their concerns, while Reviewer SUww maintained a reservation about the lack of sufficient evidence for the claim that the method “learns modality-consistent implicit auxiliary representations for missing-modality inference”. After carefully reading the submission, the reviews, and the discussion, the reviewers reached a general consensus that the paper addresses a practically meaningful and realistic problem in low-light video enhancement and presents solid empirical results under modality-missing conditions. Although the analysis of modality-consistent implicit representations could be further strengthened, this weakness does not substantially undermine the overall contribution of the paper.

The reconmendation is: Accept.